# Inference Scaling of LLM Ensembling: Bridging Token Spaces with Token Translation

## Abstract

Large language models (LLMs) exhibit diverse strengths and weaknesses across tasks, motivating recent efforts to ensemble multiple models to harness their complementary capabilities, boosting test-time performance. While model diversity and capability are known to influence ensemble effectiveness, a persistent challenge in LLM ensembling arises from mismatched tokenizer vocabularies. Existing alignment strategies typically rely on token-level embeddings or string-level heuristics of tokens, overlooking the tokenizer priors embedded during LLM pretraining. Specifically, tokenizers such as Byte-Pair Encoding (BPE) and Unigram are constructed by statistically analyzing large pretraining corpora to identify frequent subword units, and they tokenize text using greedy or probabilistic algorithms that reflect these learned subword distributions. In this work, we propose a novel and remarkably simple *Token Translation* (ToT) method that explicitly leverages these tokenizer priors to bridge heterogeneous token spaces. Our method is lightweight, requiring only a few lines of code, pre-computable, and highly efficient at inference. To further enhance robustness, we incorporate token-level model uncertainty to dynamically reweight each model's contribution during decoding. Extensive evaluations across diverse model combinations and tasks demonstrate that our method consistently outperforms existing ensembling baselines.

## 1 Introduction

Large language models (LLMs) (Touvron et al., 2023; OpenAI, 2023) have demonstrated strong performance across tasks such as question answering (Kamalloo et al., 2023), summarization (Zhang et al., 2024), and reasoning (Jaech et al., 2024; Guo et al., 2025). Owing to differences in data, architectures, and training objectives, LLMs exhibit complementary strengths and weaknesses (Jiang et al., 2023b; Yao et al., 2025). This motivates *test-time ensembling*, a practical approach to combine multiple pretrained LLMs for improved performance and adaptability without additional fine-tuning.

Test-time ensembling methods generally fall into two categories: response-level (Feng et al., 2024; Jiang et al., 2023b; Lu et al., 2023) and logit-level (or token-level) ensembling (Yao et al., 2025; Yu et al., 2024; Huang et al., 2024; Phan et al., 2025). Response-level approaches select among complete outputs or route inputs to a model, offering a simple but coarse way to combine LLMs. Logit-level approaches instead operate directly on token (or sub-token) distributions during generation, enabling finer-grained integration of model predictions. However, they face a key challenge: *tokenizer mismatch*, as models often adopt heterogeneous tokenization schemes and vocabularies, making the alignment of token-level predictions nontrivial.

LLMs adopt distinct tokenization schemes (e.g., BPE (Sennrich et al., 2015), Unigram (Kudo, 2018)), each trained on a separate pretraining corpus. These schemes differ in segmentation granularity, vocabulary, and subword decomposition, resulting in heterogeneous token spaces. Such mismatches prevent direct alignment of token-level distributions across models and can lead to sparse or unstable mappings, especially when vocabularies only partially overlap.

Prior works attempted to align tokens across models by comparing token embedding similarities (Huang et al., 2024) or using string-level heuristics such as string distance (Wan et al., 2024a). They fail due to inconsistent token boundaries and lack of grounded semantic correspondence. However, these strategies often overlook a critical source of information: *tokenizer priors*—the statistical

Table 1: Comparisons of LLM ensemble methods. **Left panel**: Comparison across five methodological dimensions. **Right panel**: Radar plot comparing ToT with the two base LLMs and the strongest ensemble baseline across six benchmarks.

| Method | Mismatch Handling | Minimal Overhead | Tokenizer Priors | Uncertainty Aware | Bidirectional |
|---|---|---|---|---|---|
| LLM-Blender (Jiang et al., 2023b) | ✗ | ✗ | ✗ | ✗ | ✗ |
| GAC (Yu et al., 2024) | ✗ | ✓ | ✗ | ✗ | ✗ |
| DeepEn (Huang et al., 2024) | ✓ | ✗ | ✗ | ✗ | ✗ |
| CES (Phan et al., 2025) | ✓ | ✗ | ✗ | ✗ | ✗ |
| UniTE (Yao et al., 2025) | ✓ | ✓ | ✗ | ✗ | ✗ |
| **ToT (Ours)** | ✓ | ✓ | ✓ | ✓ | ✓ |

patterns captured during tokenizer training on large corpora. Widely deployed tokenizers like BPE and Unigram segment text by greedily selecting high-frequency subword units, and this behavior implicitly encodes valuable knowledge about how linguistic input is represented in each LLM.

To address the challenge of tokenizer mismatch in token-level LLM ensembling, we propose a simple and highly effective alignment strategy that leverages *tokenizer priors*, i.e., the statistical patterns implicitly captured by tokenizers during training on large corpora. Our key idea is to align heterogeneous token spaces through an operation we term ***token translation***: for any token from a source model, we first decode it into its original text form using the source tokenizer, and then re-encode this text using the target model's tokenizer. This reveals how the target tokenizer interprets the same content and provides a grounded correspondence between token spaces. Intuitively, it approximates the token sequence the target model is likely to generate when expressing the same underlying content, capturing both interpretation and generation preferences. To enhance alignment quality, we further introduce a bidirectional translation scheme that integrates alignments in both directions to capture complementary structures. This approach is easy to implement, pre-computable for efficiency, and enables robust token-level fusion across models with diverse vocabularies. Detailed comparisons of existing methods are provided in Table 1, with the left panel showing the method-level characteristics and the right panel reporting quantitative performance. As highlighted, our proposed ToT method consistently checks all the desired properties and achieves the best overall results.

After aligning token predictions into a shared space, we address the challenge of aggregating outputs from models with varying reliability through an ***uncertainty-aware ensembling*** mechanism. In practice, different models may exhibit varying levels of confidence depending on the context, and treating them equally can dilute stronger signals. To address this, we dynamically reweigh each model's token distribution based on prediction entropy: models with lower uncertainty get greater influence, while uncertain ones are downweighted. This adaptive weighting allows the ensemble to emphasize more reliable token-level signals and improves reliability across diverse inputs.

We validate our method through extensive experiments across multiple model combinations, and downstream tasks. Our results show that the proposed approach consistently outperforms existing LLM ensembling methods in terms of accuracy. These results further demonstrate that ensembling performance improves as more capable LLMs are incorporated, under optimal configurations using our proposed method. Our main contributions are summarized as follows:

- We highlight the untapped potential of tokenizer priors for bridging heterogeneous token spaces and addressing tokenizer mismatch, a key bottleneck in token-level LLM ensembling.

- We introduce a bidirectional token alignment method ToT based on *token translation*, a simple yet effective operation that aligns tokens across models by leveraging their tokenizer behaviors.

- We propose an uncertainty-aware ensembling strategy that adaptively reweighs model contributions at each token step based on prediction confidence, enhancing output quality.

- We demonstrate that our method consistently improves performance across diverse tasks and model combinations, achieving an average absolute gain of 5.95 points, offering a scalable, well-founded solution for training-free LLM ensembling.

## 2 RELATED WORKS

Ensembling large language models (LLMs) has emerged as a promising direction for improving prediction accuracy, robustness, and calibration. Existing methods can be broadly grouped into three

categories: *response-level*, *logit-level*, and *training-time* ensembling. Our work focuses on token-level fusion and introduces a lightweight *token translation* strategy to address tokenizer mismatch while preserving fine-grained generation behavior.

**Response-level ensembling** aggregates complete outputs from different models. Early methods such as PairRanker (Jiang et al., 2023b) and routing-based ensembles (Feng et al., 2024; Lu et al., 2023) select the best response based on reranking or input-conditioned selection. More recent techniques like SweetSpan (Xu et al., 2024a) attempt to fuse spans across responses. While conceptually simple, these approaches ignore token-level information and often struggle to generalize across tasks and model combinations.

**Logit-level ensembling** operates at a finer granularity by aggregating token logits during generation. We use this term broadly to include variants that operate on tokens, sub-tokens (bytes), or token spans. UniTE (Yao et al., 2025) uses top-$k$ token union for alignment, while GAC (Yu et al., 2024) and DeePen (Huang et al., 2024) rely on dense mapping matrices between vocabularies to perform projection-based fusion. Span-level ensembling (Xu et al., 2025) merges consecutive tokens into larger segments, but substantially increases computation cost. Sub-token-level ensembling (Gu et al., 2024; Phan et al., 2025) such as CES, instead decomposes tokens into smaller units, but suffers from a lack of tokenizer priors, often requiring expensive search. Our method avoids these issues by leveraging tokenizer priors to construct sparse, grounded token mappings via a translation mechanism.

**Training-time ensembling** focuses on distilling multiple model outputs into a student model. FuseLLM (Wan et al., 2024a) and FuseChat (Wan et al., 2024b) perform logit-based distillation, while EVA (Xu et al., 2024b) and EnsW2S (Agrawal et al., 2024) learn vocabulary projection layers to integrate predictions. Ent (Ruan et al., 2022) also explore weighted ensembling, but their weights are learned during training, unlike our test-time aggregation of frozen heterogeneous LLMs. While effective for model compression, these approaches require model retraining and full access to parameters, making them inapplicable in test-time scenarios where models are frozen or proprietary.

## 3 METHODOLOGY

In this section, we present our test-time ensembling framework for token-level fusion across LLMs with heterogeneous tokenizers. We first define the *token alignment* problem arising from vocabulary mismatch, then introduce *token translation* to align token spaces, along with a *bidirectional* variant for added robustness. Finally, we describe an *uncertainty-aware* ensembling strategy that reweights model outputs by confidence. An overview is shown in Figure 1.

### 3.1 PROBLEM FORMULATION: TOKEN ALIGNMENT ACROSS HETEROGENEOUS TOKENIZERS

Token-level ensembling across LLMs requires the ability to align token distributions across models with heterogeneous tokenizers. We follow the setting adopted by prior work such as UniTE (Yao et al., 2025), GAC (Yu et al., 2024), and DeePen (Huang et al., 2024), where a designated *main model* serves as the reference, and one or more *assist models* provide auxiliary signals. At each decoding step, each model outputs a probability distribution over its own vocabulary: $\boldsymbol{p}^{(\text{main})} \in \Delta^{|\mathcal{V}_{\text{main}}|}$ of tokenizer $\texttt{Tok}_{\text{main}}$ and $\boldsymbol{p}^{(\text{assist})} \in \Delta^{|\mathcal{V}_{\text{assist}}|}$ of tokenizer $\texttt{Tok}_{\text{assist}}$, where $\Delta^{|\mathcal{V}|}$ denotes the probability simplex, i.e., the set of non-negative vectors in $\mathbb{R}^{|\mathcal{V}|}$ that sum to one.

The goal is to construct a *mapping matrix* $\boldsymbol{M} \in \mathbb{R}^{|\mathcal{V}_{\text{assist}}| \times |\mathcal{V}_{\text{main}}|}$ for each assist model, where $\boldsymbol{M}_{i,j}$ indicates the alignment between assist token $v_i^{(\text{assist})}$ and main token $v_j^{(\text{main})}$. By performing row-normalization on $\boldsymbol{M}$, we obtain the transition matrix $\boldsymbol{T} \in \mathbb{R}^{|\mathcal{V}_{\text{assist}}| \times |\mathcal{V}_{\text{main}}|}$, where $\boldsymbol{T}_{i,j} = \Pr(v_j^{(\text{main})}|v_i^{(\text{assist})})$ is the probability of transiting assist token $v_i^{(\text{assist})}$ to main token $v_j^{(\text{main})}$. Using $\boldsymbol{T}$, the assist model's probability is projected into the main vocabulary space via $\boldsymbol{T}^\top \boldsymbol{p}^{(\text{assist})}$. Finally, these aligned distributions are ensembled by their linear combination.

### 3.2 TOKEN TRANSLATION VIA DECODE → ENCODE

Given the problem setting above, the key challenge lies in constructing a meaningful and efficient transition matrix $\boldsymbol{T}$, as the vocabularies $\mathcal{V}_{\text{main}}$ and $\mathcal{V}_{\text{assist}}$ from different LLM families are often

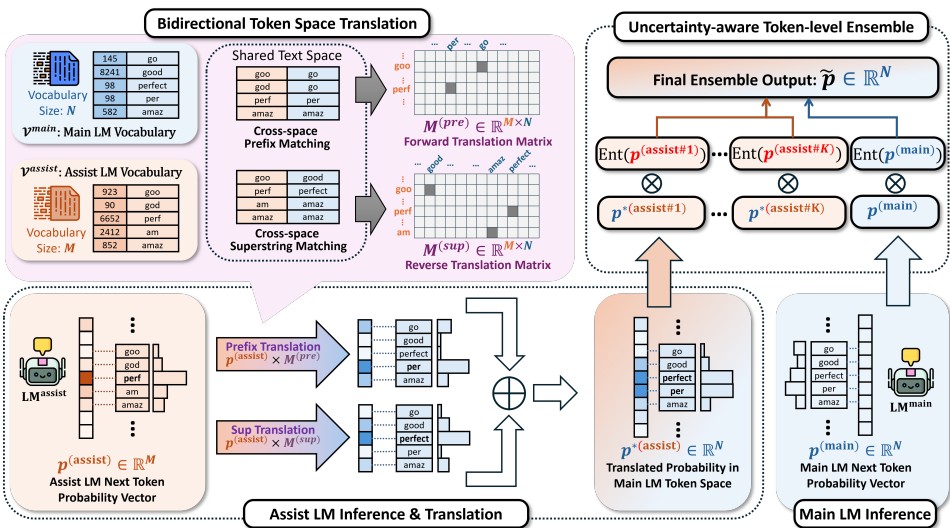

Figure 1: Illustration of our **ToT** framework for token-level ensembling across heterogeneous LLMs. Assist model predictions are translated into the main model's token space via bidirectional token translation using tokenizer priors. Final predictions are fused using uncertainty-aware weighting for robust ensemble decoding.

only partially overlapping. Traditional approaches based on string matching or embedding similarity frequently fail due to inconsistent token boundaries and the absence of semantic correspondence. To address this, we propose an alignment strategy based on *tokenizer priors*, which leverages the statistical segmentation behavior inherently learned during tokenizer training, and further introduce a simple yet effective mechanism, *token translation*, that uses these priors to construct $T$ and establish semantic correspondences across heterogeneous vocabularies.

Tokenizers such as BPE and Unigram are trained on large corpora to segment text into frequent and meaningful subword units. These statistical segmentation behaviors encode valuable priors about how each tokenizer decomposes natural language. Unlike model weights that are optimized for downstream objectives, tokenizer behavior is fixed and deterministic once trained. Therefore, it provides a stable and semantically grounded signal for reasoning about how different models represent and generate language.

Previous alignment strategies often rely on post-hoc heuristics such as string similarity or embedding proximity. However, such approaches fail to capture generation preferences intrinsic to each tokenizer. Our key insight is that tokenizer segmentation behavior have already encoded a deterministic causal structure for next-token generation, and leveraging these priors leads to more reliable cross-model alignment. A formal justification and analysis are provided in Appendix C.

---

**Example: Failure of Similarity-based Alignment**

Suppose the assist tokenizer produces the token good, while the main tokenizer includes go, goo, and nice. The intended continuation is goods. A surface-similarity heuristic might align good to goo or even nice due to lexical or semantic resemblance. However, this ignores how the main tokenizer actually segments and generates: in practice it may begin with go, then continue with ods (e.g., go+od+s). Mapping good to goo or nice biases decoding toward irrelevant continuations such as goose or nicely, deviating from the desired goods. By respecting tokenizer priors, token translation aligns good to the main prefix go, preserving the ideal generative priors.

---

**Method.** To account for such discrepancies, we propose to align tokens through their tokenizer-internal decoding preferences. Specifically, for assist token id $i$, we **decode** it into its raw text form $v_i^{(\text{assist})}$ using the assist tokenizer. We then re-**encode** $v_i^{(\text{assist})}$ using the main tokenizer to obtain a sequence $[t_{i,1}^{(\text{main})}, \ldots, t_{i,m}^{(\text{main})}]$ of token ids from $\mathcal{V}_{\text{main}}$. We map the assist token id $i$ to the first token

id $t_{i,1}^{(\text{main})}$ in this sequence:

$$M_{i,j} = \begin{cases} 1 & \text{if } t_{i,1}^{(\text{main})} = j, \\ 0 & \text{otherwise.} \end{cases}$$

Intuitively, this token captures the initial prefix the main model is most likely to generate for the given content, providing a generation-compatible alignment.

This procedure is repeated for all tokens in $\mathcal{V}_{\text{assist}}$, yielding a sparse mapping matrix $M \in \mathbb{R}^{|\mathcal{V}_{\text{assist}}| \times |\mathcal{V}_{\text{main}}|}$. Each row has a single nonzero entry: $M_{i,j} = 1$ if $v_i^{(\text{assist})}$ aligns with $v_j^{(\text{main})}$. This sparsity makes the alignment efficient to store and apply, and it reflects a semantically informed, generation-aware projection from the assist to the main vocabulary spaces.

## 3.3 BIDIRECTIONAL TRANSLATION FOR COMPLEMENTARY ALIGNMENT

While unidirectional token translation offers a simple and effective means of token alignment, it might be limited by the asymmetry. In particular, it may favor shorter prefixes or overlook longer and semantically richer tokenizations that arise in the reverse direction. To enhance alignment robustness and capture complementary structures, we propose a *bidirectional translation* scheme.

The key idea is to consider both translation directions between the assist and main vocabularies and combine the outcomes. Specifically, in addition to the forward mapping $M^{(\text{pre})}$ where each assist token is mapped to the first token generated by the main tokenizer (as done in Section 3.2), we also obtain a reverse mapping $M^{(\text{sup})}$ where each main token is decoded and then re-tokenized using the assist tokenizer. Intuitively, $M^{(\text{pre})}$ maps the assist token to its *prefix* main token, while $M^{(\text{sup})}$ maps the assist token to its *superstring* main tokens. These complementary mappings are further fused to yield a more robust bidirectional alignment.

Formally, we use $\text{Pre}(v_i^{(\text{assist})})$ to denote the mapped prefix of $v_i^{(\text{assist})}$ and $\text{Sup}(v_i^{(\text{assist})})$ to denote the set of mapped superstrings of $v_i^{(\text{assist})}$. Let $v_i^{(\text{assist})} \in \mathcal{V}_{\text{assist}}$ and $v_j^{(\text{main})} \in \mathcal{V}_{\text{main}}$ be tokens from the assist and main vocabularies, respectively. Define the forward mapping matrix $M^{(\text{pre})}$ such that $M_{i,j}^{(\text{pre})} = 1$ if $v_j^{(\text{main})} = \text{Pre}(v_i^{(\text{assist})})$ under token translation. Similarly, define the reverse mapping matrix $M^{(\text{sup})}$ such that $M_{i,j}^{(\text{sup})} = 1$ if $v_j^{(\text{main})} \in \text{Sup}(v_i^{(\text{assist})})$ under reverse token translation. Instead of taking the maximum of the two, we apply a weighted combination:

$$M^{(\text{assist} \to \text{main})} = M^{(\text{pre})} + \alpha \cdot M^{(\text{sup})},$$

where $\alpha \in [0, 1]$ is a tunable weight that controls the influence of reverse translation. We set $\alpha < 1$ to prioritize prefix translation, ensuring alignment remains consistent with the main model's generative behavior, while still allowing reverse direction signals to contribute additional coverage for longer or nested token patterns.

This strategy reflects the intuition that forward translation maintains the semantics most compatible with the main model's own vocabulary, while reverse translation can introduce complementary patterns that improve the recall. This is particularly useful when the reverse tokenizer favors generating longer subwords that align with semantic units absent in the forward match.

To generate a valid transition matrix, the fused mapping matrix is further row-normalized to produce a transition matrix $T^{(\text{assist} \to \text{main})}$, that is

$$T^{(\text{assist} \to \text{main})} = \text{diag}(M^{(\text{assist} \to \text{main})} \mathbf{1})^{-1} M^{(\text{assist} \to \text{main})}.$$

Based on the translation matrix transition matrix $T^{(\text{assist} \to \text{main})}$, we can decompose the token translation process into two parts, including translating to prefix and translating to substring, as follows

$$\begin{aligned} \boldsymbol{p}^{(\text{assist} \to \text{main})}(i) &= \left( T^{(\text{assist} \to \text{main})\top} \boldsymbol{p}^{(\text{assist})} \right)(i) \\ &= \sum_{v_j^{(\text{assist})}} \Pr(\text{Pre}) \Pr\left( v_i^{(\text{main})} | v_j^{(\text{assist})}, \text{Pre} \right) \Pr\left( v_j^{(\text{assist})} \right) + \sum_{v_j^{(\text{assist})}} \Pr(\text{Sup}) \Pr\left( v_i^{(\text{main})} | v_j^{(\text{assist})}, \text{Sup} \right) \Pr\left( v_j^{(\text{assist})} \right) \\ &= \sum_j \Big\{ \underbrace{\frac{1}{1 + \alpha N}}_{\Pr(\text{Pre})} \underbrace{\mathbb{1}\left( v_i^{(\text{main})} = \text{Pre}(v_j^{(\text{assist})}) \right)}_{\Pr\left( v_i^{(\text{main})} | v_j^{(\text{assist})}, \text{Pre} \right)} \underbrace{\boldsymbol{p}^{(\text{assist})}(j)}_{\Pr\left( v_j^{(\text{assist})} \right)} + \underbrace{\frac{\alpha N}{1 + \alpha N}}_{\Pr(\text{Sup})} \underbrace{\text{Unif}\left( \mathbb{1}\left( v_i^{(\text{main})} \in \text{Sup}(v_j^{(\text{assist})}) \right) \right)}_{\Pr\left( v_i^{(\text{main})} | v_j^{(\text{assist})}, \text{Sup} \right)} \underbrace{\boldsymbol{p}^{(\text{assist})}(j)}_{\Pr\left( v_j^{(\text{assist})} \right)} \Big\}, \end{aligned}$$

---

**Algorithm 1** Token Translation (ToT)

---

**Input:** Tokenizers $\mathtt{Tok}_{\text{assist}}, \mathtt{Tok}_{\text{main}}$; Vocabularies $\mathcal{V}_{\text{assist}}, \mathcal{V}_{\text{main}}$; weight $\alpha$
**Initialize:** $\boldsymbol{M}^{(\text{pre})} \leftarrow \boldsymbol{0}_{|\mathcal{V}_{\text{assist}}| \times |\mathcal{V}_{\text{main}}|}, \boldsymbol{M}^{(\text{sup})} \leftarrow \boldsymbol{0}_{|\mathcal{V}_{\text{assist}}| \times |\mathcal{V}_{\text{main}}|}$
**for** $i = 1, 2, ..., |\mathcal{V}_{\text{assist}}|$
$\quad v_i^{(\text{assist})} \leftarrow \mathtt{Tok}_{\text{assist}}.\text{decode}(i)$
$\quad [t_1, \dots, t_m] \leftarrow \mathtt{Tok}_{\text{main}}.\text{encode}(v_i^{(\text{assist})})$
$\quad \boldsymbol{M}_{i,t_1}^{(\text{pre})} \leftarrow 1$
**for** $j \in 1, 2, ..., |\mathcal{V}_{\text{main}}|$
$\quad v_j^{(\text{main})} \leftarrow \mathtt{Tok}_{\text{main}}.\text{decode}(j)$
$\quad [t_1', \dots, t_n'] \leftarrow \mathtt{Tok}_{\text{assist}}.\text{encode}(v_j^{(\text{main})})$
$\quad \boldsymbol{M}_{t_1',j}^{(\text{sup})} \leftarrow 1$
$\boldsymbol{M}^{(\text{assist} \rightarrow \text{main})} \leftarrow \boldsymbol{M}^{(\text{pre})} + \alpha \cdot \boldsymbol{M}^{(\text{sup})}$
**Return:** $\boldsymbol{M}^{(\text{assist} \rightarrow \text{main})}$

---

where $N = |\mathbb{1}(v_i^{(\text{main})} \in \mathtt{Sup}(v_j^{(\text{assist})}))|$ is the number of target tokens that are the prefix of $v_i^{(\text{main})}$. The above equation provides insights on the translation process: with probability $\Pr(\mathtt{Pre}) = \frac{1}{1+\alpha N}$, the probability mass on assist token is transited to the mapped prefix main token, and with probability $\Pr(\mathtt{Sup}) = \frac{\alpha N}{1+\alpha N}$, the probability mass on assist token is uniformly transited to the mapped superstring main tokens.

### 3.4 Uncertainty-Aware Token-Level Ensembling

After establishing a shared alignment space, we ensemble token-level predictions from multiple LLMs with an *uncertainty-aware fusion strategy*.

Different models exhibit varying levels of confidence depending on the input. Treating all models equally may amplify noise or disproportionately emphasize low-confidence predictions. To address this, we use entropy-based weighting, following observations from UniTE (Yao et al., 2025) that low-confidence tokens tend to introduce noise. To improve robustness, we only compute entropy over the top-$k$ tokens in each model's predicted distribution (Ma et al., 2025), filtering out uninformative tails.

Specifically, define $\text{Top}_k(\boldsymbol{p}^{(\text{assist})})$ as the top-$k$ highest scoring assist tokens. The uncertainty-based weight $w_{\text{ent}}^{(\text{assist})}$ is computed as the inverse entropy of the Top-$k$ tokens, that is

$$w_{\text{ent}}^{(\text{assist})} = \frac{1}{H(\boldsymbol{p}^{(\text{assist})})}, \quad \text{where } H(\boldsymbol{p}^{(\text{assist})}) = - \sum_{v \in \text{Top}_k(\boldsymbol{p}^{(\text{assist})})} \frac{\boldsymbol{p}^{(\text{assist})}(v)}{Z} \log \frac{\boldsymbol{p}^{(\text{assist})}(v)}{Z},$$

where $Z = \sum_{v \in \text{Top}_k(\boldsymbol{p}^{(\text{assist})})} \boldsymbol{p}^{(\text{assist})}(v)$. A similar procedure can be applied to the main model, denoted $w_{\text{ent}}^{(\text{main})}$. We then normalize all weights across the main model and all assist models so that they sum to one. The final ensembled distribution over the main model's vocabulary is:

$$\tilde{\boldsymbol{p}} = w^{(\text{main})} \cdot \boldsymbol{p}^{(\text{main})} + \sum_{\kappa=1}^{K} w^{(\text{assist}\#\kappa)} \cdot \boldsymbol{T}^{(\text{assist}\#\kappa)^\top} \boldsymbol{p}^{(\text{assist}\#\kappa)},$$

where $[w^{(\text{main})}, w^{(\text{assist}\#1)}, \cdots w^{(\text{assist}\#K)}] \in \Delta^{K+1}$ are the normalized uncertainty-based weights that control the contribution of the main model and assist models. This formulation maintains alignment with the main model's prediction while allowing assist models to refine uncertain or ambiguous cases when they are confident.

All token-translation matrices are precomputed and cached as highly sparse matrices. Each column (corresponding to a single assist-model token) has only one or a small number of non-zero entries that point to its top-ranked aligned tokens in the main model's vocabulary. This sparsity enables very fast lookup and efficient memory usage.

## 4 Experiments

In this section, we design experiments to answer the following research questions:

Table 2: Evaluation results across datasets for base models and LLM ensemble methods in two-model setting. Ensembling uses **LLaMA-3.1-8B-Instruct** (light blue) as the main model and **InternLM-2.5-7B-Chat** (light pink) as the assist model. Best results are in **bold**, and second-best are underlined.

| Model | NQ | TriviaQA | ARC-c | MMLU | GSM8K | PIQA | Avg |
|---|---|---|---|---|---|---|---|
| *Base Models* | | | | | | | |
| Gemma-2-9B-IT | 13.49 | 52.53 | 49.57 | 41.56 | 78.23 | 61.96 | 49.22 |
| GLM-4-9B-Chat | 24.27 | 58.94 | 88.06 | 64.68 | 76.41 | 82.58 | 65.82 |
| InternLM-2.5-7B-Chat | 26.62 | 63.23 | 85.13 | 70.57 | 83.93 | 87.47 | 69.49 |
| LLaMA-3.1-8B-Instruct | 27.51 | 65.57 | 78.72 | 67.38 | 82.87 | 82.86 | 67.82 |
| Mistral-7B-Instruct | 25.37 | 66.61 | 73.42 | 58.76 | 61.74 | 69.55 | 59.91 |
| Qwen2.5-7B-Instruct | 15.04 | 53.23 | 87.35 | 70.46 | 78.67 | 85.25 | 65.67 |
| Yi-1.5-9B-Chat | 14.54 | 51.74 | 80.64 | 66.27 | 64.69 | 80.64 | 59.75 |
| *LLM Ensemble Methods* | | | | | | | |
| LLM-Blender | 25.10 | 61.12 | 76.01 | 65.89 | 80.15 | 80.43 | 64.78 |
| GAC | 22.86 | 65.45 | 70.83 | 68.92 | 75.62 | 78.90 | 63.10 |
| CES | 30.54 | 66.20 | 81.80 | 68.10 | 81.10 | 87.72 | 69.24 |
| DeepEn | 28.63 | 66.32 | 75.49 | 68.23 | 81.33 | 79.24 | 66.21 |
| Unite | 29.11 | 67.45 | 83.23 | 69.56 | 84.43 | 86.63 | 70.07 |
| **ToT (Ours)** | **32.30** | **72.58** | **85.74** | **71.89** | **87.41** | **88.69** | **73.77** |
| *Improve over Base LLM* | (+4.79) | (+7.01) | (+7.02) | (+4.51) | (+4.54) | (+5.83) | (+5.95) |
| Oracle (Roofline) | 39.11 | 78.53 | 89.06 | 80.48 | 91.20 | 94.90 | 78.88 |

- **RQ1**: Does our ensemble framework consistently improve the performance of base models across diverse tasks?

- **RQ2**: How do different components, such as uncertainty modeling and prefix/superstring-based token mappings, and alignment strategies contribute to ensemble effectiveness?

- **RQ3**: How efficient is our method in terms of preprocessing overhead, inference latency, and GPU memory compared to existing ensemble baselines?

- **RQ4**: How does performance scale with the number and capacity of involved models?

## 4.1 EXPERIMENTAL SETUP

**Models.** We conduct all experiments using a diverse set of open-source chat and instruct-tuned models, including `LLaMA-3.1-8B-Instruct` (Grattafiori et al., 2024), `InternLM-2.5-7B/20B-Chat` (Team, 2023), `GLM-4-9B-Chat` (GLM et al., 2024), `Mistral-7B-Instruct-v0.3` (Jiang et al., 2023a), `Qwen2.5-7B-Instruct` (Yang et al., 2024), `Yi-1.5-9B-Chat` (Young et al., 2024), and `Gemma-2-9B-IT` (Team et al., 2024). These models reflect a range of widely adopted architectures spanning practical model sizes and are among the most recent publicly available versions at the time of the experiments.

**Baselines.** We compare against four representative test-time ensembling methods. **LLM-Blender** (Jiang et al., 2023b) uses a reward model (`PairRanker`) and a fusion model (`GenFuser`) to rerank and merge responses. Due to over-generation issues with `GenFuser`, we use only the reward-based selection. **GAC** (Yu et al., 2024) maps token probabilities into a unified space using a learned matrix and performs token-level aggregation. **CES** (Phan et al., 2025) ensembles probabilities at the sub-token (byte) level for fill-in-the-middle tasks. **DeepEn** (Huang et al., 2024) projects model outputs into a shared latent space based on overlapping vocabularies, using relative representation theory. **UniTe** (Yao et al., 2025) uses a top-$k$ union strategy to align and filter tokens.

**Benchmarks.** We evaluate nine benchmarks across four task categories. **(1) Comprehensive exams:** `MMLU` (5-shot) (Hendrycks et al., 2020) and `ARC-C` (0-shot) (Clark et al., 2018), assessing subject knowledge and natural science reasoning. **(2) Reasoning:** `GSM8K` (Cobbe et al., 2021) (4-shot with chain-of-thought prompting (Wei et al., 2022)), `PIQA` (0-shot) (Bisk et al., 2020), and `HumanEval` (0-shot) (Chen et al., 2021), covering arithmetic, commonsense, and program synthesis. **(3) Knowledge-intensive QA:** `TriviaQA` (5-shot) (Joshi et al., 2017) and `NaturalQuestions` (`NQ`) (5-shot) (Kwiatkowski et al., 2019). **(4) Summarization:** `SAMSum` (0-shot) (Gliwa et al., 2019), a dialogue summarization dataset. Dataset details are in Appendix A.

**Experimental Setup.** We evaluate both two-model and multi-model ensembling across the afore-mentioned benchmarks. For the two-model setting, we select `LLaMA-3.1-8B-Instruct` and `InternLM-2.5-7B-Chat` as the main pair, chosen for their strong and comparable performance. For multi-model settings, we include `Qwen2.5-7B-Instruct` and additional models to assess scalability. We also explore diverse model pairings to test generality. Unless otherwise specified, we fix the hyperparameters to $\alpha = 0.5$ and set the top-$k$ tokens for uncertainty estimation to $k = 50$. All experiments are conducted on NVIDIA A100 80GB GPUs.

## 4.2 MAIN RESULTS (RQ1)

As shown in Table 2, our method (ToT) achieves the best performance across all benchmarks, with an average improvement of +5.95 points over the main model (`LLaMA-3.1-8B-Instruct`). It delivers consistent gains across tasks (e.g., +4.79 on `NQ`, +7.01 on `TriviaQA`, +7.02 on `ARC-C`), and remains robust even when the main model underperforms, for example, surpassing both base models on `ARC-C` (85.74 vs. 78.72 and 85.13).

We also report a *roofline accuracy*, the percentage of examples answered correctly by at least one base model, as an empirical upper bound for training-free ensembles. ToT closely tracks this upper bound across tasks, especially on reasoning-oriented benchmarks (GSM8K and ARC-c), indicating that it effectively aggregates complementary reasoning signals at inference. The remaining headroom is concentrated in knowledge-intensive settings, suggesting that integrating external knowledge or task-specific priors could further narrow the gap. Compared to prior methods, `LLM-Blender` relies on coarse-grained output ranking and shows limited improvement. `Unite` uses top-$k$ token selection but lacks generation-awareness. Other methods like `GAC` and `DeepEn` suffer from poor token alignment across vocabularies. These results highlight the value of tokenizer-aware alignment and uncertainty modeling in test-time LLM ensembling.

## 4.3 ABLATION STUDY (RQ2))

We perform ablation experiments on the NQ benchmark to assess the contribution of each module and the effectiveness of alternative alignment strategies (Table 3). Our two-model ToT ensemble improves the main model by +4.79 points (32.30), and adding a third model (Mistral-7B-Instruct) further boosts performance to 32.94, demonstrating scalability. Among core components, removing uncertainty weighting yields a minor drop (32.11), indicating stability benefits. In contrast, removing the prefix-based alignment matrix $M^{\text{pre}}$ significantly reduces performance (30.75), confirming the importance of generation-aware alignment. Omitting the reverse mapping $M^{\text{sup}}$ (31.86) or the clipping factor $\beta$ (32.12) also leads to degradation, highlighting the utility of bidirectional alignment and confidence balancing. We also compare ToT with alternative alignment strategies, including semantic similarity (27.63), lexical matching (28.55), and edit-distance-based alignment (Wan et al., 2024a) (29.12). All underperform our method, underscoring the benefit of leveraging tokenizer priors over surface heuristics.

Table 3: Ablation study on NQ. We compare ToT with different variants and token mapping strategies.

| Setting | NQ Acc. |
|---|---|
| *Baselines* | |
| InternLM-2.5-7B-Chat | 26.62 |
| LLaMA-3.1-8B-Instruct | 27.51 |
| **ToT** (2 models) | **32.30** |
| +Mistral-7B (3 models) | **32.94** |
| *Ablations* | |
| w/o Uncertainty | 32.11 |
| w/o $M^{\text{pre}}$ | 30.75 |
| w/o $M^{\text{sup}}$ | 31.86 |
| w/o $\beta$ | 32.12 |
| *Alternative Mapping Strategies* | |
| with Semantic | 27.63 |
| with Lexical | 28.55 |
| with MinED (Wan et al., 2024a) | 29.12 |

Hyperparameter studies for $\alpha$ - $k$ (Appendix B.1) show stable performance across wide ranges, with fixed defaults near-optimal and consistently outperforming baselines. Results across diverse model combinations (Appendix B.2) demonstrate gains that persist when swapping the *main* and *assist* roles; the improvement magnitude correlates with the assist model's strength. Token-alignment visualizations (Appendix B.3) reveal sparse, semantically grounded mappings and cases where the fused distribution corrects errors even when both base models' predictions are wrong. Appendix B.4 presents empirical studies on tokenizer alignment properties, scalability across models and languages, and the effect of alternative uncertainty measures that further confirm our approach's effectiveness.

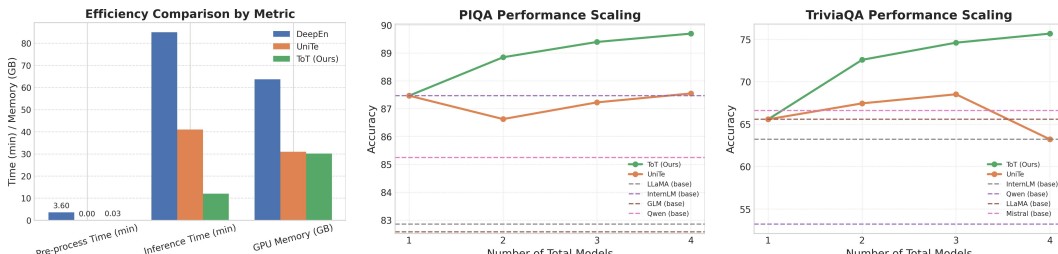

(a) Time & memory cost across methods (lower is better).

(b) Performance scaling on PIQA (ToT vs. UniTE).

(c) Performance scaling on Trivi-aQA (ToT vs. UniTE).

Figure 2: Efficiency and scaling overview. **Left**: wall-clock time and peak memory across ensembling methods. **Middle/Right**: scaling behavior of ToT vs. UniTE across base models.

## 4.4 EFFICIENCY ANALYSIS (RQ3)

Figure 2a compares the time and memory costs of different ensembling methods. ToT achieves clear efficiency advantages in both preprocessing and inference. Unlike DeepEn, which requires costly offline alignment, ToT relies only on lightweight `decode-encode` operations to construct a sparse mapping (0.03 min vs. 3.6 min). At inference, ToT performs a single sparse matrix multiplication without altering model architecture or decoding logic, yielding substantially lower time (12 min vs. 85 min) and memory use (30.2 GB vs. 63.7 GB). Compared to UniTE, ToT is also more efficient due to its sparse and fixed-tokenizer design. Overall, ToT delivers strong accuracy (Table 2) while maintaining practical speed and memory scalability.

## 4.5 SCALABILITY ACROSS MODEL SIZES AND QUANTITIES (RQ4)

To evaluate scalability, we consider two settings: (1) ensembling models of different scales, and (2) increasing the number of ensembled models.

As shown in Table 4, ToT achieves substantial gains when applied to large-scale models. By ensembling `InternLM2.5-20B-Chat` with `LLaMA-3.1-8B-Instruct`, ToT consistently outperforms both base models across

Table 4: Ensemble performance across model scales on four tasks.

| Model | HumanEval | SAMSum | NQ | PIQA |
|---|---|---|---|---|
| LLaMA-3.1-8B-Instruct | 73.17 | 31.70 | 27.51 | 82.86 |
| InternLM2.5-20B-Chat | 79.73 | 32.29 | 28.53 | 88.19 |
| UniTe | 78.43 | 32.45 | 31.65 | 88.96 |
| **ToT (Ours)** | **90.24** | **32.92** | **34.82** | **89.79** |

all four tasks (e.g., +10.51 on `HumanEval`, +7.31 on `NQ`). This highlights the flexibility of our approach in handling model pairs with heterogeneous capacities and confirms its effectiveness in large-model regimes.

Furthermore, Figure 2b-2c illustrates performance scaling trends as more models are added to the ensemble. For both `TriviaQA` and `PIQA`, ToT demonstrates a clear upward trajectory, achieving steady improvements from 2 to 4 models. In contrast, UniTE's performance plateaus or even degrades when additional models are introduced, likely due to its less robust token selection strategy and lack of generation preference modeling. This further confirms ToT's robustness and generality when scaling across both model diversity and ensemble size.

## 5 CONCLUSION

We propose a lightweight and effective framework ToT for token-level ensembling of LLMs, addressing the core challenge of tokenizer mismatch. Our method aligns heterogeneous token spaces via a simple *token translation* based on tokenizer priors, enhanced by bidirectional alignment and uncertainty-aware fusion. Extensive experiments across tasks, model sizes, and ensemble settings show that ToT consistently outperforms existing baselines while maintaining high efficiency. Looking ahead, a key challenge is selecting complementary model combinations. As the LLM landscape grows, developing adaptive strategies for model pairing and ensemble configuration will be essential for maximizing performance and diversity.

**Ethical Statement.** This work focuses on algorithmic methods for test-time ensembling of publicly available large language models. No human subjects or sensitive personal data are involved. All datasets used in our experiments (e.g., MMLU, ARC-C, GSM8K, TriviaQA, NaturalQuestions, SAMSum, HumanEval, PIQA) are standard public benchmarks widely adopted in prior research. We have taken care to follow the ICLR Code of Ethics by ensuring appropriate dataset usage, acknowledging limitations, and avoiding any harmful applications. The proposed method is purely methodological and does not pose direct risks related to privacy, security, or fairness.

**Reproducibility Statement.** We provide detailed descriptions of our methodology in Section 3, including the construction of token translation matrices, bidirectional alignment, and uncertainty-aware weighting. Experimental settings, datasets, and evaluation protocols are specified in Section 4 and Appendix A. Ablation studies and hyperparameter analyses are presented in Section 4.3 and Appendix B to demonstrate robustness. All token translation matrices are precomputable and lightweight. To further support reproducibility, we've incorporated code, precomputed mappings, and experiment scripts in the supplementary material.

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

# Appendix

## CONTENTS

# A  DATASET STATISTICS

We evaluate nine benchmark datasets across four representative task categories to assess language model performance under diverse reasoning and comprehension requirements. **(1) Comprehensive Exams:** MMLU (5-shot, 5,000 test examples) (Hendrycks et al., 2020) is a multiple-choice benchmark that spans 57 diverse subjects across STEM, humanities, social sciences, and more. Each question contains four answer choices, and models are evaluated using in-context few-shot prompting and accuracy as the metric. ARC-Challenge (0-shot, 1,172 examples) (Clark et al., 2018) is a science reasoning benchmark targeting difficult grade-school level questions with multiple-choice formats. Models must directly select the correct answer without in-context examples. **(2) Reasoning:** GSM8K (4-shot with chain-of-thought prompting (Wei et al., 2022), 1,319 examples) (Cobbe et al., 2021) consists of grade-school math word problems. The model is prompted with four examples that explicitly demonstrate step-by-step intermediate reasoning, and the final prediction is considered correct only if the numeric answer matches the gold label exactly. PIQA (0-shot, 1,838 examples) (Bisk et al., 2020) evaluates physical commonsense reasoning: models are given a naturalistic question involving physical interactions and must choose between two plausible outcomes. HumanEval (0-shot, 164 examples) (Chen et al., 2021) is a code generation benchmark where each input specifies a function signature and docstring in Python, and the model is expected to synthesize correct code. Performance is evaluated using pass@1, indicating the percentage of problems solved correctly in a single attempt. **(3) Knowledge-Intensive QA:** TriviaQA (5-shot, 11,313 examples) (Joshi et al., 2017) contains open-domain factoid questions curated from trivia websites, with Wikipedia-based ground truth answers. Each input includes five QA exemplars followed by a new question; accuracy is measured by exact match. NaturalQuestions (NQ) (5-shot, 3,610 examples) (Kwiatkowski et al., 2019) consists of real anonymized Google search queries, paired with short answers derived from Wikipedia articles. We follow prior work in using short-form extractive answer strings and compute exact match as the evaluation metric. **(4) Summarization:** SAMSum (0-shot, 818 examples) (Gliwa et al., 2019) is a single-document summarization task where the input is a multi-turn dialogue between fictitious participants, and the model must produce a concise and coherent summary. No demonstrations are provided, and performance is measured using ROUGE-L.

Following prior work (Yao et al., 2025; Huang et al., 2024), we construct the prompt by providing the question followed by the answer format, optionally including in-context examples and chain-of-thought prompting using the phrase "let's think step-by-step" (Wei et al., 2022).

# B  FURTHER ABLATION STUDY AND VISUALIZATION

In this section, we first perform a comprehensive hyperparameter study to demonstrate the robustness of our method under various settings. We then conduct an ablation study using different pairs of large language models (LLMs) in the ensemble to further verify its robustness across diverse model combinations. Finally, we present representative output examples and token alignment visualizations to provide qualitative insights into the behavior and effectiveness of our approach.

## B.1  HYPER-PARAMETER STUDY

Figures 3, 4, and 5 demonstrate robustness across the key hyperparameters $\alpha$, $\beta$, and $k$. Here, $\alpha$ controls the blend between prefix and superstring mappings when aligning token sequences; $\beta$ controls the *strength of the main model* by enforcing a minimum contribution—implemented as clipping the normalized weight so that $w^{(\text{main})} \leftarrow \max\big(\beta,\, w^{(\text{main})}\big)$ and renormalizing the remaining mass across assist models; and $k$ sets how many (top) tokens are used to compute model uncertainty. Despite their different roles, all three are easy to configure and yield stable performance over wide ranges. Across settings, our method consistently outperforms the base models, making it practical without tuning; thus we fix these values uniformly in all experiments.

## B.2  ABLATION STUDY ON VARIOUS MODEL ENSEMBLING

Figure 6 provides a comprehensive view of the performance gains achieved when ensembling different main and assist model pairs. Two key observations emerge.

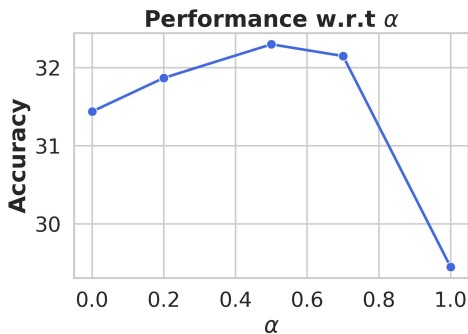

Figure 3: Performance with respect to $\alpha$, representing the ratio between prefix and superstring mappings on the NQ dataset with InternLM and LLaMA models.

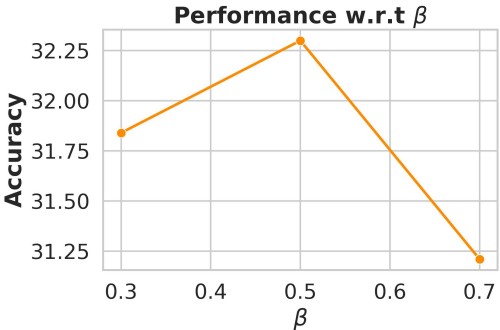

Figure 4: Performance with respect to $\beta$, denoting the minimum contribution of the main model to the final prediction, evaluated on the NQ dataset with InternLM and LLaMA.

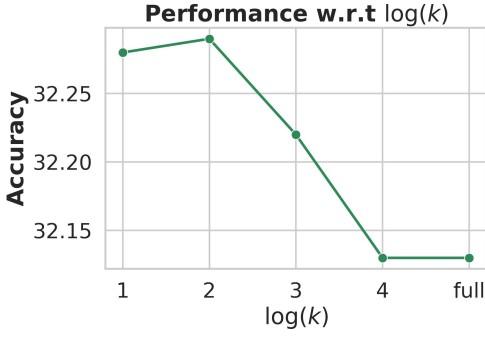

Figure 5: Performance with respect to $\log k$, where $k$ is the number of tokens used to compute uncertainty, evaluated on the NQ dataset using InternLM and LLaMA.

First, our method is highly robust, as it consistently delivers substantial performance improvements across all combinations of main and assist models. No matter which model is used as the main model—GLM, InternLM, LLaMA, or Mistral—the introduction of an assist model yields a notable gain over the base performance. This demonstrates that our ensembling strategy generalizes well and is not overly sensitive to the choice of models involved.

Second, the magnitude of the gain tends to correlate with the strength of the assist model. Specifically, using stronger models like InternLM and LLaMA as assist models leads to the largest improvements,

with performance gains exceeding 5 points in some cases. For example, pairing InternLM as the assist model with GLM as the main model yields the highest observed gain. This suggests that stronger assist models provide more informative or complementary signals during ensembling, thus boosting the overall prediction quality.

Together, these results validate the versatility and effectiveness of our method, showing it works well in diverse model settings and particularly excels when high-capacity assist models are available.

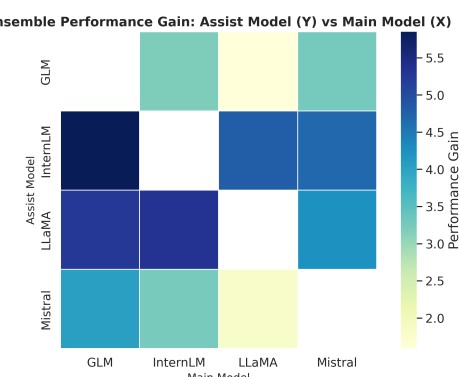

Figure 6: Performance gain with different main and assist models.

## B.3 DEMONSTRATION OF ALIGNMENT AND ENSEMBLING

Table 5 compares three token mapping methods: string similarity, embedding similarity, and our proposed Token Translation. String similarity is based on edit distance at the string level, while embedding similarity measures cosine similarity between token embeddings. In contrast, Token Translation (using prefix-based mapping) leverages both the model's generation preferences and tokenizer priors to identify mappings that align more closely with actual generation behavior. As evidenced by the performance results in Table 3, both string- and embedding-based methods often produce suboptimal or irrelevant mappings—for example, mapping "Papa" to "Luna" or "Administrator" to a Chinese token—because they overlook how tokens function during decoding. In contrast, Token Translation consistently produces compact, semantically coherent segments such as "Admin" and "P" that better match the tokenizer's segmentation and generation dynamics. This demonstrates the effectiveness of our approach in generating mappings that are not only linguistically and structurally aligned, but also more suitable for downstream ensemble generation.

Table 5: Comparison of Token Mapping Methods

| Method | Source Token | Mapped Token |
|---|---|---|
| **String sim** | Administrator | Administr |
| **Embedding sim** | Administrator | Administrator in Chinese |
| **Token Translation** | Administrator | Admin |
| **String sim** | _CAT | _CA |
| **Embedding sim** | _CAT | _VAR |
| **Token Translation** | _CAT | _C |
| **String sim** | Papa | Pap |
| **Embedding sim** | Papa | Luna |
| **Token Translation** | Papa | P |

Next, we present output examples of our ensemble method in Tables 6 and 7, which not only correct errors made by the base models but also outperform baseline methods on the GSM8K dataset. Correct answers are highlighted in green, while incorrect ones are marked in red. These cases clearly demonstrate the effectiveness of our method in enhancing reasoning accuracy and robustness by leveraging complementary strengths of different models.

Table 6: Case study of ToT and other ensembling methods on GSM8K.

| Question | Blake and Kelly are having a contest to see who can run the most in 15 minutes. They decide to do it on a football field that is 100 yards long. Blake runs back and forth 15 times. Kelly runs back and forth once, and then decides that she doesn't want to run next to Blake, so she starts to run to the 40-yard line and back. She does this 34 times. How much farther does the winner run than the loser? |
|---|---|
| **LLaMA** | Blake runs 15 times back and forth on a 100-yard field, so he runs 15*2*100=3000 yards. Kelly runs 34 times to the 40-yard line and back, so she runs 34*2*40=2720 yards. Blake runs 3000 yards and Kelly runs 2720 yards, so Blake runs 3000-2720=280 yards more than Kelly. 

 The answer is **280** |
| **InternLM** | First, let's calculate how far Blake runs. A football field is 100 yards long, and he runs back and forth 15 times. 
 So, Blake runs 100 yards * 15 = 1500 yards. 
 Next, let's calculate how far Kelly runs. She runs back and forth once, which is 100 yards * 2 = 200 yards. 
 Then, she starts running to the 40-yard line and back, which is 40 yards * 2 = 80 yards. She does this 34 times, so she runs an additional 80 yards * 34 = 2720 yards. 
 In total, Kelly runs 200 yards + 2720 yards = 2920 yards. 
 To find out how much farther the winner runs than the loser, we subtract the distance that Blake ran from the distance that Kelly ran: 2920 yards - 1500 yards = 1420 yards. 

 The answer is **1420** |
| **UniTE** | Blake and Kelly are having a running contest on a 100-yard football field. Blake runs back and forth 15 times, and since each round trip is 100 yards, he runs a total of $15 \times 100 = 1500$ yards. 
 Kelly first runs one full trip across the field and back, which is $2 \times 100 = 200$ yards. 
 After that, she runs to the 40-yard line and back 34 times. 
 Since the 40-yard line is not the full length, we compute her additional distance as $40 \times 34 = 1360$ yards. 
 Adding this to her initial 200 yards, Kelly runs a total of 1560 yards. 
 Comparing the two, Kelly runs $1560 - 1500 = 60$ yards more than Blake. 

 The answer is **60** |
| **ToT (Ours)** | Blake runs back and forth 15 times on a 100-yard field, so he runs 15*2*100=3000 yards. 
 Kelly runs back and forth once, which is 2*100=200 yards. 
 Then she runs to the 40-yard line and back, which is 2*40=80 yards. 
 She does this 34 times, so she runs 34*80=2720 yards. 
 The total distance that Kelly runs is 200+2720=2920 yards. 
 The difference between the distances that Blake and Kelly run is 3000-2920=80 yards. 

 The answer is **80** |

Table 7: Case study of ToT and other ensembling methods on GSM8K.

| Question | John buys milk for 2 dollars, eggs for 3 dollars, light bulbs for 3 dollars, cups for 3 dollars, and roach traps for 4 dollars. If there is a 10% tax on all nonfood items how much did everything cost? |
|---|---|
| LLaMA | John buys milk for 2 dollars, eggs for 3 dollars, and roach traps for 4 dollars, so the total cost of the food items is 2+3+4=9 dollars. 
 The nonfood items are the light bulbs, cups, and roach traps. The total cost of the nonfood items is 3+3+4=10 dollars. 
 There is a 10% tax on the nonfood items, so the tax is 10% of 10 dollars, which is 0.1*10=1 dollar. 
 The total cost of the nonfood items with tax is 10+1=11 dollars. 
 The total cost of everything is the cost of the food items plus the cost of the nonfood items with tax, so the total cost is 9+11= 20 |
| InternLM | First, let's calculate the cost of the non-food items before tax. The non-food items are the light bulbs, cups, and roach traps. 
 The cost of the light bulbs is \$3, the cups are also \$3, and the roach traps are \$4. 
 So the total cost of the non-food items is \$3 + \$3 + \$4 = \$10. 
 The answer is 10 |
| UniTE | John buys milk for 2 dollars, eggs for 3 dollars, and roach traps for 4 dollars, so the total for food items is $2 + 3 + 4 = 9$ dollars. 
 The nonfood items include light bulbs, cups, and roach traps, totaling $3 + 3 + 4 = 10$ dollars. 
 A 10% tax applies to nonfood items, so the tax is 10% of 10 dollars, which is $0.1 \times 10 = 1$ dollar. 
 Adding the tax, the nonfood total becomes $10 + 1 = 11$ dollars. 
 Combining the food and taxed nonfood totals, the final amount is $9 + 11 =$ 20 |
| ToT (Ours) | First, let's calculate the total cost of the non-food items. The non-food items are the light bulbs, cups, and roach traps. 
 The cost of the light bulbs is \$3, the cups are \$3, and the roach traps are \$4. 
 So the total cost of the non-food items is \$3 + \$3 + \$4 = \$10. 
 Next, we need to calculate the tax on the non-food items. The tax rate is 10%, so we need to find 10% of \$10. 
 To do this, we multiply \$10 by 0.10 (which is the decimal equivalent of 10%). 
 So the tax on the non-food items is \$10 * 0.10 = \$1. 
 Now, we can add the tax to the cost of the non-food items to find the total cost of the non-food items. 
 The total cost of the non-food items is \$10 + \$1 = \$11. 
 Finally, we need to add the cost of the milk and eggs to the total cost of the non-food items to find the total cost of everything. 
 The cost of the milk is \$2 and the cost of the eggs is \$3. 
 So the total cost of everything is \$2 + \$3 + \$11 = 16 |

### B.4 EMPIRICAL ANALYSIS OF TOKENIZERS AND PERFORMANCE

To complement the formal analysis in Appendix C, we present an empirical study of tokenizer behavior and its effect on cross-model alignment. This analysis examines how heterogeneous vocabularies interact in practice and how these structural properties influence ensemble performance. We begin by measuring token alignment coverage across representative LLM families.

#### B.4.1 TOKEN ALIGNMENT COVERAGE ACROSS MODELS

We begin by quantifying token-level alignment coverage across five representative LLM families. Table 8 reports the percentage of tokens in one vocabulary that find direct alignment matches in another. Coverage varies substantially across model pairs (e.g., 85.43% for LLaMA3→Qwen2.5 vs. 36.30% for GLM4→Mistral), revealing the strong tokenizer-induced structural differences that motivate a deterministic translation mechanism such as ToT.

Table 8: Cross-token alignment coverage among representative LLM tokenizers.

| Model Pair | internlm2 (92k) | Qwen2.5 (152k) | LLaMA3 (128k) | GLM4 (151k) | Mistral (32k) |
|---|---|---|---|---|---|
| internlm2 (92k) | – | 61.51% | 62.70% | 54.75% | 79.65% |
| Qwen2.5 (152k) | 61.51% | – | 85.43% | 33.05% | 75.57% |
| LLaMA3 (128k) | 62.70% | 85.43% | – | 39.15% | 75.79% |
| GLM4 (151k) | 54.75% | 33.05% | 39.15% | – | 36.30% |
| Mistral (32k) | 79.65% | 75.57% | 75.79% | 36.30% | – |

#### B.4.2 FORWARD/BACKWARD MAPPING PRECISION

Next, we examine how many non-shared tokens can be mapped via ToT. Table 9 reports results when mapping several assist models into the internlm2 vocabulary. Because vocabularies differ greatly in size and segmentation granularity, forward (assist→main) and backward (main→assist) mapping coverage can be highly asymmetric—for example, Qwen2.5→internlm2 achieves 91.35% forward coverage but only 23.09% backward coverage. Smaller vocabularies (e.g., Mistral's 32k) also show the opposite pattern. These results illustrate why alignment would be better to consider tokenizer directionality rather than assume symmetric similarity.

Table 9: Forward (assist→main) and backward (main→assist) mapping coverage into internlm2.

| Model → internlm2 | Direction | Mapped | Non-Shared | Coverage |
|---|---|---|---|---|
| Qwen2.5 (152k) | Forward | 32,540 | 35,616 | 0.9135 |
|  | Backward | 21,969 | 95,136 | 0.2309 |
| LLaMA3 (128k) | Forward | 17,004 | 34,521 | 0.4927 |
|  | Backward | 6,283 | 70,233 | 0.0894 |
| GLM4 (151k) | Forward | 37,705 | 41,876 | 0.9006 |
|  | Backward | 32,371 | 100,884 | 0.3210 |
| Mistral (32k) | Forward | 1,061 | 66,443 | 0.0159 |
|  | Backward | 5,719 | 6,667 | 0.8576 |

#### B.4.3 SEMANTIC COHERENCE ANALYSIS

To evaluate whether ToT's alignments are semantically meaningful, Table 10 reports the average BGE embedding (Xiao et al., 2023) similarity between paired tokens. Scores are consistently high (0.70–0.97), confirming semantic coherence while remaining below the theoretical maximum, indicating that alignment is still governed by tokenizer, induced structural decomposition rather than semantic nearest neighbors.

#### B.4.4 SCALING TO LARGER BACKBONES

To assess scalability, we evaluate ToT on Mixtral-8×7B and Qwen2.5-32B-Instruct. Table 11 shows that ToT consistently improves over both individual models and the UniTe baseline, demonstrating that the alignment benefits persist even for strong dense and MoE backbones.

Table 10: Semantic similarity of aligned tokens using BGE embeddings.

| Model → internlm2 | Direction | Avg Sim | Max Avg Sim |
|---|---|---|---|
| Qwen2.5 (152k) | Forward | 0.7778 | 0.8347 |
| | Backward | 0.9407 | 0.9904 |
| LLaMA3 (128k) | Forward | 0.6999 | 0.7969 |
| | Backward | 0.8863 | 0.9880 |
| GLM4 (151k) | Forward | 0.7818 | 0.8398 |
| | Backward | 0.9669 | 0.9959 |
| Mistral (32k) | Forward | 0.7142 | 0.7679 |
| | Backward | 0.8411 | 0.9508 |

Table 11: Scalability evaluation on larger backbones.

| Model | SAMSum | NQ | Avg |
|---|---|---|---|
| Mixtral-8×7B | 32.8 | 30.9 | 31.9 |
| Qwen2.5-32B-Instruct | 34.1 | 33.7 | 33.9 |
| UniTe (baseline) | 34.0 | 34.5 | 34.3 |
| **ToT (Ours)** | **35.3** | **36.9** | **36.1** |

### B.4.5 MULTILINGUAL EVALUATION: FLORES EN→DE AND EN→ZH

We further test robustness on FLORES En→De and En→Zh translation. Table 12 shows that ToT improves over both single models and UniTe, particularly in En→Zh where tokenization differences are largest.

Table 12: Multilingual evaluation on FLORES subsets.

| Dataset | Model | BLEU (↑) |
|---|---|---|
| **En→De** | LLaMA | 29.87 |
| | InternLM | 26.74 |
| | UniTe | 33.17 |
| | **Ours** | **36.13** |
| **En→Chinese** | LLaMA | 27.45 |
| | InternLM | 32.10 |
| | UniTe | 34.82 |
| | **Ours** | **37.56** |

### B.4.6 UNCERTAINTY MEASURE COMPARISON

Finally, we compare several uncertainty metrics within the same routing framework. As shown in Table 13, top-$k$ entropy achieves the strongest average performance. This aligns with the observation that long-tail logits introduce noise, while the top-$k$ region contains the meaningful decision mass for autoregressive decoding.

## C FORMAL ANALYSIS OF TOKEN TRANSLATION

In this section, we provide a rigorous justification for our Token Translation (ToT) framework. We show that ToT is not a heuristic approximation, but a deterministic implementation of *Exact Causal Alignment* between heterogeneous token spaces. Under standard tokenizer assumptions, ToT recovers the ground-truth conditional probability required for cross-model alignment.

### C.1 PROBLEM SETUP

Let $\mathcal{V}_{\text{assist}}$ and $\mathcal{V}_{\text{main}}$ denote the vocabularies of the assist and main models, respectively. We posit a shared, continuous text space $\mathcal{S}$ representing all possible raw text strings. The alignment objective is

Table 13: Comparison of uncertainty measures for routing.

| Uncertainty Measure | Description | Avg Score ($\uparrow$) |
|---|---|---|
| Entropy | Full-distribution entropy | 32.13 |
| Variance | Variance of the distribution | 31.10 |
| Calibration | $1 - \max(p)$ confidence proxy | 31.56 |
| Disagreement | $\mathrm{KL}(\text{main}\|\text{assist})$ | 31.28 |
| **Top-$k$ Entropy (Ours)** | Entropy over normalized top-$k$ mass | **32.30** |

to compute

$$P\Big(v^{(\text{main})} \mid v^{(\text{assist})}\Big), \qquad v^{(\text{assist})} \in \mathcal{V}_{\text{assist}}, \ v^{(\text{main})} \in \mathcal{V}_{\text{main}}.$$

The true conditional probability is

$$P\Big(v^{(\text{main})} \mid v^{(\text{assist})}\Big) = \sum_{s \in \mathcal{S}} \underbrace{P\Big(v^{(\text{main})} \mid s\Big)}_{\text{Main Prior}} \cdot \underbrace{P\Big(s \mid v^{(\text{assist})}\Big)}_{\text{Assist Decoding}}. \tag{1}$$

## C.2 Tokenizer Assumptions

Modern BPE and Unigram tokenizers satisfy the following assumptions.

**Assumption 1 (Deterministic Decoding).** Each assist token $v^{(\text{assist})}$ deterministically decodes to a unique raw string:

$$P(s \mid v^{(\text{assist})}) = \mathbb{1}\Big[s = \text{Tok}_{\text{assist}}.\text{decode}(v^{(\text{assist})})\Big].$$

**Assumption 2 (Canonical Encoding Prior).** For any string $s$, the main tokenizer produces a unique canonical sequence $\text{Tok}_{\text{main}}.\text{encode}(s) = [t_1, t_2, \dots]$. Due to autoregressive causality,

$$P\Big(v^{(\text{main})} \mid s\Big) = \mathbb{1}\Big[v^{(\text{main})} = \text{First}(\text{Tok}_{\text{main}}.\text{encode}(s))\Big].$$

## C.3 Causal Alignment

Under Assumptions 1–2, the intractable sum in Eq. 1 collapses to

$$P_{\text{true}}\Big(v^{(\text{main})} \mid v^{(\text{assist})}\Big) = \mathbb{1}\Big[v^{(\text{main})} = \text{First}\Big(\text{Tok}_{\text{main}}.\text{encode}\Big(\text{Tok}_{\text{assist}}.\text{decode}(v^{(\text{assist})})\Big)\Big)\Big].$$

This is the exact causal mapping from assist-token space to main-token space.

## C.4 ToT Recovers the Exact Alignment

ToT constructs a forward translation matrix $M^{(\text{pre})}$:

$$\widehat{P}_{\text{ToT}}\Big(v^{(\text{main})} \mid v^{(\text{assist})}\Big) \triangleq M^{(\text{pre})}_{v^{(\text{assist})}, v^{(\text{main})}},$$

where

$$M^{(\text{pre})}_{i,j} = \mathbb{1}[j = \text{First}(\text{Tok}_{\text{main}}.\text{encode}(\text{Tok}_{\text{assist}}.\text{decode}(i)))].$$

Thus,

$$\mathbb{E}\Big[\widehat{P}_{\text{ToT}}\Big] = P_{\text{true}},$$

showing that ToT exactly recovers the deterministic causal alignment induced by the tokenizers.

### C.5 CAUSAL VALIDITY

Consider an assist token that expands to a multi-token main sequence $[t_1, t_2, \ldots, t_n]$. Define:

- $E_{\text{seq}}$: the event that the main model generates the full sequence;
- $E_{t_1}$: the event that it generates $t_1$.

Since

$$E_{\text{seq}} \subseteq E_{t_1} \quad \implies \quad P(E_{\text{seq}}) \leq P(E_{t_1}),$$

the first token $t_1$ is a necessary causal precursor for generating the entire sequence. If the assist model assigns probability $p$ to $v^{(\text{assist})}$, then the main model must assign at least $p$ to $t_1$. Therefore,

$$\widehat{P}(t_1) \leftarrow p$$

preserves probability mass and yields a deterministic causal mapping for next-token prediction.

As shown in Table 3, this prefix-only mapping already surpasses strong baselines. Moreover, the backward translation matrix $M^{(\text{sup})}$ restores the residual probability mass spread across longer sequences, providing improved robustness through bidirectional alignment.

## D USE OF LARGE LANGUAGE MODELS

During the preparation of this paper, we made limited use of large language models (LLMs), specifically ChatGPT, as an auxiliary writing tool. The LLM was used exclusively for stylistic refinement, including improvements to fluency, grammar, and readability of text initially drafted by the authors. All scientific content, including problem formulation, methodology, experiments, analyses, and overall narrative, was entirely conceived and validated by the authors. Thus, the role of LLMs was restricted to text polishing and does not constitute authorship.

## E LIMITATIONS

Our approach assumes access to model tokenizers and output probabilities, which may not be available for all proprietary APIs. Additionally, selecting optimal model combinations for ensembling remains an open challenge.

