# OpenReview forum: "Inference Scaling of LLM Ensembling: Bridging Token Spaces with Token Translation"
_ICLR.cc/2026/Conference — Submitted to ICLR 2026_

### Official Review · Reviewer_TweT · 2025-10-30

**Soundness:** 3
**Presentation:** 3
**Contribution:** 2
**Rating:** 4
**Confidence:** 4

**Summary:**

This paper introduces a new cross-tokenizer ensemble approach. The method constructs a mapping matrix that aligns one tokenizer’s vocabulary to another (base) vocabulary using prefix relations. In addition, the authors propose an entropy-based weighting scheme to combine ensemble outputs adaptively. Experimental results demonstrate consistent improvements over previous methods.

**Strengths:**

While the method is largely heuristic, its reported performance appears reasonable and may be of some practical interest. The use of tokenizer priors is a sensible design choice and appears more effective in practice compared to earlier methods such as Unite and DeepEN.

The experimental evaluation is sufficiently thorough to support the authors’ main claims.

The proposed method is more efficient than prior works such as DeepEN.

**Weaknesses:**

The token-translation method is mainly heuristics and is not theoretically grounded compared to other approach, for example the character-ensemble baseline converts the every model to the same base character alphabet with statistical guarantees.

For cross-tokenizer problem, it is necessary to include the amount of overlapping tokens between the vocabularies. For example, Llama3 and Qwen2.5 shares a large amount of similar tokens.

The idea of using entropy for model combination is not new (see [1], for example). It would be important to include a comparison against other methods with entropy-weighted ensembles. In particular, Table 2 could include a separate row showing the proposed approach without entropy weighting (i.e., “ToT without entropy weighting”) to clarify its contribution.


[1] Ruan, Yangjun, et al. "Weighted Ensemble Self-Supervised Learning." The Eleventh International Conference on Learning Representations.

**Questions:**

Line 122-123, what do you mean by "grounded semantics correspondence"? This was never defined nor rigorously stated in the paper.

How does the selection of the target tokenizer for mapping (in an ensemble setup) influence overall performance?

Also see weakness.

---

> ### Author Response · Authors · 2025-11-21
> **Response to Reviewer TweT (Part 1/3)**
>
> Dear Reviewer TweT,
>
> Thank you for your valuable feedback, which has helped us improve the clarity and quality of our paper. We have incorporated the corresponding revisions into the updated manuscript, where all revisions and additions are highlighted in blue for your convenience. We detail our responses below, and please kindly let us know if our replies address your concerns.
>
> > **W1.** The token-translation method is mainly heuristics and is not theoretically grounded compared to other approach.
>
> Thank you for the question. Although Token Translation is simple to implement and requires no tuning after a single one-time mapping, **it is not a heuristic method**. To clarify, we present a formal justification showing that ToT implements a **deterministic form of causal alignment** between heterogeneous token probability spaces. Under standard assumptions satisfied by modern BPE and Unigram tokenizers, ToT produces a **deterministic causal mapping** of the conditional probabilities involved in cross-model alignment. This provides a clear **theoretical foundation**.
>
> Let $\mathcal{V}\_{\text{assist}}$ and $\mathcal{V}\_{\text{main}}$ denote the vocabularies of the assist and main models, respectively. We posit a shared, continuous ground-truth text space $\mathcal{S}$ representing all possible raw text strings. The objective of token alignment is to project the probability mass from an assist token $v^{(\text{assist})} \in \mathcal{V}\_{\text{assist}}$ to a target main token $v^{(\text{main})} \in \mathcal{V}\_{\text{main}}$.
>
> The true conditional probability is:
>
> $$
> P(v^{(\text{main})} \mid v^{(\text{assist})})=
> \sum\_{s \in \mathcal{S}}
> \underbrace{P(v^{(\text{main})} \mid s)}\_{\text{Main Prior}} \cdot
> \underbrace{P(s \mid v^{(\text{assist})})}\_{\text{Assist Decoding}}
> \tag{1}
> $$
>
> To make the above tractable, we introduce two assumptions satisfied by modern BPE/Unigram tokenizers:
>
> - **Assumption 1 (Deterministic Decoding).**
>   A token $v^{(\text{assist})}$ maps deterministically to a unique textual form string. That is, $P(s \mid v^{(\text{assist})})$ follows a Dirac delta distribution:
>
>   $$
>   P(s \mid v^{(\text{assist})})
>   = \mathbb{1}[s = Tok\_{\text{assist}}.\mathrm{decode}(v^{(\text{assist})}) \]
>   $$
>
> - **Assumption 2 (Canonical Encoding Prior).**
>   Given any string $s$, the main model’s tokenizer deterministically produces a unique canonical tokenization:
>
>   $$Tok\_{\text{main}}.\mathrm{encode}(s) = [t\_1, t\_2, \dots]$$
>
>   Because autoregressive models generate text in a causal left-to-right manner, any generation of $s$ must begin with its first token $t\_1$. Accordingly, in our alignment derivation we use the indicator:
>
>   $$\mathbb{1}[v^{(\text{main})} = t\_1]$$
>
>   which reflects the deterministic **tokenization mapping**, rather than any assumption about the determinism of the model’s next-token **probability distribution**.
>
>
> Under these assumptions, the intractable sum in Eq. (1) collapses into a single indicator function, which serves as our Ground Truth Causal Alignment:
>
> $$
> P\_{\text{true}}(v^{(\text{main})} \mid v^{(\text{assist})})=
> \mathbb{1}\left[
> v^{(\text{main})}=
> \mathrm{First}\left(
> Tok\_{\text{main}}.\mathrm{encode}(Tok\_{\text{assist}}.\mathrm{decode}(v^{(\text{assist})}))
> \right)\right]
> $$
>
> Our ToT method computes exactly this alignment via the forward translation matrix $M^{(\text{pre})}$:
>
> $$
> \hat{P}\_{\text{ToT}}(v^{(\text{main})} \mid v^{(\text{assist})})
> \triangleq
> M^{(\text{pre})}\_{v^{(\text{assist})}, v^{(\text{main})}}
> $$
>
> By construction:
>
> $$
> M^{(\text{pre})}\_{i,j} = 1
> \quad\text{iff}\quad
> j = \mathrm{First}(Tok\_{\text{main}}.\mathrm{encode}(Tok\_{\text{assist}}.\mathrm{decode}(i)))
> $$
>
> Thus:
>
> $$
> \mathbb{E}[\hat{P}\_{\text{ToT}}] = P\_{\text{true}}
> $$
>
> This proves that ToT is a **deterministic causal mapping** (and, in deterministic tokenizers, an exact calculator) of the underlying causal conditional probability.
>
>
> Consider the case where a single assist token expands to a multi-token main sequence $[t\_1, t\_2, \dots, t\_n]$.
>
> Let:
>
> - $E\_{\text{seq}}$ = the event that the main model generates the full sequence
> - $E\_{t\_1}$ = the event that it generates $t\_1$
>
> Since:
>
> $$
> E\_{\text{seq}} \subseteq E\_{t\_1}
> \quad\Longrightarrow\quad
> P(E\_{\text{seq}}) \le P(E\_{t\_1})
> $$
>
> If the assist model predicts $v^{(\text{assist})}$ with probability $p$, then the main model must assign at least probability $p$ to $t\_1$, because $t\_1$ is the **necessary causal precursor** of the full token sequence.
>
> Therefore assigning:
>
> $$
> \hat{P}(t\_1) \leftarrow p
> $$
>
> faithfully preserves probability mass and yields a **deterministic causal mapping** for next-token prediction.
>
> The performance of the prefix alone is already impressive and surpasses the baselines, as shown in Table 3 in the paper. Furthermore, through back translation, $M^{(\text{sup})}$ complements the probability mass, allowing the ensemble to better capture the representation of the underlying sequence.

---

> ### Author Response · Authors · 2025-11-21
> **Response to Reviewer TweT (Part 2/3)**
>
> > **W2.** For cross-tokenizer problem, it is necessary to include the amount of overlapping tokens between the vocabularies. For example, Llama3 and Qwen2.5 shares a large amount of similar tokens.
>
> Thanks for the question. We report the **percentage of aligned token pairs** between five representative LLMs, reflecting the **coverage of direct cross-token mappings**
>
> | Model Pairs                     | internlm2_5-7b-chat (92k) | Qwen2.5-7B-Instruct (152k) | Llama-3.1-8B-Instruct (128k) | glm-4-9b-chat (151k) | Mistral-7B-Instruct-v0.3 (32k) |
> |-------------------------------------|---------------------------|-----------------------------|-------------------------------|-----------------------|---------------------------------|
> | internlm2_5-7b-chat (92k)          | —                         | 61.51%                      | 62.70%                        | 54.75%                | 79.65%                          |
> | Qwen2.5-7B-Instruct (152k)         | 61.51%                    | —                           | 85.43%                        | 33.05%                | 75.57%                          |
> | Llama-3.1-8B-Instruct (128k)       | 62.70%                    | 85.43%                      | —                             | 39.15%                | 75.79%                          |
> | glm-4-9b-chat (151k)               | 54.75%                    | 33.05%                      | 39.15%                        | —                     | 36.30%                          |
> | Mistral-7B-Instruct-v0.3 (32k)     | 79.65%                    | 75.57%                      | 75.79%                        | 36.30%                | —                               |
>
>
> ---
>
> > **W3.** The idea of using entropy for model combination is not new (see [1], for example). It would be important to include a comparison against other methods with entropy-weighted ensembles. In particular, Table 2 could include a separate row showing the proposed approach without entropy weighting (i.e., “ToT without entropy weighting”) to clarify its contribution.
>
> Thank you for raising this point. We already include the **ablation “ToT without uncertainty”** in Table 3, which isolates the contribution of the uncertainty component. We have also cited [1] for a comprehensive discussion in the revised paper.
>
> To evaluate whether other uncertainty measures could be competitive, we implemented several alternatives, all defined on the full probability distribution for fairness and consistency. These include **full-distribution entropy**, **distribution variance**, **calibration-based confidence using 1−max(p)**, and a **main–assist disagreement score** based on the KL divergence between the two models’ predicted distributions. In the disagreement-based version, **strong divergence** between main and assist decreases the assist’s weight, while **agreement increases it**. We replaced only the uncertainty scoring function and kept the routing framework identical across all methods. Empirically, **top-k entropy achieves the highest average performance (32.30)**, while the full-distribution measures and the disagreement-based score yield slightly lower results. The comparison is summarized in the table below.
>
>
> | Uncertainty Measure                    | How It Works                                   | Avg. Score (↑) |
> |----------------------------------------|-------------------------------------------------|----------------|
> | Entropy              | Entropy over full vocabulary distribution      | 32.13          |
> | Variance             | Variance over distribution                | 31.10          |
> | Calibration (1 − max prob)             | Confidence proxy using max probability         | 31.56          |
> | Disagreement          | KL(main \| assist); large gap → lower assist   | 31.28          |
> | **Top-k Entropy (Ours)**               | Entropy over normalized top-k mass             | **32.30**      |
>
> These results clarify the contribution of entropy weighting and highlight that the proposed **top-k formulation provides strong overall performance**.

---

> ### Author Response · Authors · 2025-11-21
> **Response to Reviewer TweT (Part 3/3)**
>
> > **Q1.** Line 122-123, what do you mean by "grounded semantics correspondence"? This was never defined nor rigorously stated in the paper.
>
> Thank you for the clarification. We have removed the phrase **“grounded semantic correspondence”** and replaced it with the more precise description **“tokenizer priors,”** which better reflects the information our method relies on.
>
> ---
>
> > **Q2.** How does the selection of the target tokenizer for mapping (in an ensemble setup) influence overall performance?
>
> Each LLM uses its own unique and internally compatible tokenizer. Our ablation study in **Figure 6** already evaluates all valid pairwise combinations among the four models, explicitly swapping which model serves as the main (target) model and which serves as the assist (source) model. Each LLM in Figure 6 uses its own SentencePiece BPE tokenizer with a distinct vocabulary. **Every main–assist assignment shows a consistent improvement over the main model baseline.** This demonstrates that the effectiveness of ToT does not depend on a particular model ordering and remains robust under role reversal.
>
> ---
>
> **Thank you again for the thoughtful review. We have dedicated substantial effort to obtaining new results and have incorporated them into the revised paper PDF to further strengthen the work. We hope our responses adequately address your concerns, and we would be happy to discuss anything further if needed.**

---

> ### Comment · Reviewer_TweT · 2025-11-22
>
> The author need to be careful about assumption 2, because the string s will need to end with EOS symbol to make it true, i.e. the summation cannot collapse to a single indicator function.

---

> > ### Author Response · Authors · 2025-11-22
> >
> > Dear Reviewer TweT,
> >
> > Thanks for the prompt response. We appreciate the opportunity to clarify. This indicator **does not assume that the next-token distribution is deterministic**. It follows solely from the **deterministic nature of tokenizer encoding**, which assigns each string a unique canonical token sequence, a property that holds broadly across modern LLM tokenizers. In particular, we assume the determinism of the mapping:
> >
> > $$Tok\_{\text{main}}.\mathrm{encode}(s)=[t\_1, t\_2, \dots]$$
> >
> > We have revised the wording to make this clearer.
> >
> > - **Assumption 2 (Canonical Encoding Prior).**
> >   Given any string $s$, the main model’s tokenizer deterministically produces a unique canonical tokenization:
> >
> >   $$Tok\_{\text{main}}.\mathrm{encode}(s) = [t\_1, t\_2, \dots]$$
> >
> >   Because autoregressive models generate text in a causal left-to-right manner, any generation of $s$ must begin with its first token $t\_1$. Accordingly, in our alignment derivation we use the indicator:
> >
> >   $$\mathbb{1}[v^{(\text{main})} = t\_1]$$
> >
> >   which reflects the **deterministic tokenization mapping**, rather than any assumption about the model’s next-token **probability distribution**.
> >
> > We sincerely thank the reviewer for the helpful suggestions and are glad to clarify further if needed.
> >
> > Best,
> >
> > #14478 Authors

---

> ### Comment · Reviewer_TweT · 2025-11-22
>
> Sorry I do not follow your point. The expression you wrote for the conditional distribution $P(v^{\text{main}}|v^{\text{assist}})$ clearly states that it is an indicator function where it is 1.0 for the value $v^{\text{main}}$ that satisfies the condition and $0$ elsewhere, which is simply not true according to [1,2]. Maybe the author misunderstood my comment. For simplicity, let set $\mathcal{V}^{\text{assist}} $ to be the base-alphabet (i.e it returns the string). Then $P(v^{\text{main}}|v^{\text{assist}}) = P(v^{\text{main}}|s)$. Let the alphabet be ${\mathrm{a},\mathrm{b}}$ and the tokenizer vocab $\mathcal{V}^{\text{main}} = [\mathrm{a,b,aa}]$. Then let say $s=\mathrm{a}$ then there are two valid tokens that satisfies this, which is $\mathrm{aa}$ and $\mathrm{a}$ (see [1]) so the conditional cannot collapse into a single point. Your comment "any generation of s must begins with..." thus overlook this issue since there are two way to generate it, with token $\mathrm{aa}$ or token $\mathrm{a}$ which is why I suggest the inclusion of EOS in the equation, also see [2], Section 2.2 on clarification of EOS.
>
> [1] Phan, Buu, et al. "Exact Byte-Level Probabilities from Tokenized Language Models for FIM-Tasks and Model Ensembles." ICLR 2025.
>
> [2] Vieira, Tim, et al. "From language models over tokens to language models over characters.", ICML 2025.

---

> > ### Author Response · Authors · 2025-11-22
> >
> > Dear Reviewer TweT,
> >
> > Thank you very much for the additional clarification and for providing concrete examples. Your comment is extremely helpful, and we appreciate the opportunity to further explain our formulation.
> >
> > To clarify the scope of our analysis, our discussion focuses specifically on the **forward mapping**, i.e., mapping an assist token to the **prefix** produced by the main model’s tokenizer. We fully agree that **a token in one vocabulary may correspond to multiple valid token sequences in another vocabulary (e.g., `"abc"` → `["ab", "c"]` or `["a", "bc"]`)**.
> >
> > Our use of the indicator function is based on the fact that:  **when we apply the *main model’s tokenizer* to the *string form* of the assist token, the encoding is deterministic**. That is:
> >
> > **$Tok\_{\text{main}}.\mathrm{encode}(s)$**
> >
> > always returns a **unique token sequence**, and therefore its **first token is uniquely determined**. The indicator only reflects this **deterministic tokenization**.
> >
> > This is precisely where our method leverages **tokenizer priors**, i.e., the structural information built into BPE/Unigram tokenization rules. As an **empirical paper**, we utilize this deterministic mapping and then validate its effectiveness through **extensive evaluation across tasks, models, and ensemble configurations**.
> >
> > Thank you for highlighting the EOS issue and directing us to Reference [2]. This clarification is highly valuable and helps formalize the equation. We will cite the reference, add an explicit EOS discussion, and refine the notation to avoid ambiguity.
> >
> > Thank you again for the constructive clarification. We appreciate the opportunity to refine our formulation.
> >
> > Best,
> >
> > #14478 Authors

---

> > > ### Comment · Reviewer_TweT · 2025-11-25
> > >
> > > Thank you for your reply. I will take all the points and re-evaluate the paper accordingly.

---

> > > > ### Author Response · Authors · 2025-11-26
> > > >
> > > > Dear Reviewer TweT,
> > > >
> > > > Thank you for your kind message and for taking our clarifications into consideration during your re-evaluation. We truly appreciate your constructive engagement throughout the discussion. Please feel free to let us know if any further points would benefit from additional explanation.
> > > >
> > > > Best,
> > > >
> > > > #14478 Authors

---

### Official Review · Reviewer_TiHi · 2025-11-01

**Soundness:** 2
**Presentation:** 3
**Contribution:** 3
**Rating:** 4
**Confidence:** 4

**Summary:**

This paper proposes a simple, lightweight, and training-free model ensembling method called Token Translation (ToT). Instead of relying on computationally expensive or semantically weak alignment methods (like embedding similarity or string heuristics), ToT leverages the "tokenizer priors", i.e., the statistical segmentation behavior learned by each model's tokenizer. Extensive experiments demonstrates the effects of this method.

**Strengths:**

S1: The authors propose a simple yet highly effective training-free ensembling method that aligns the vocabularies of different models using only tokenization rules, presenting substantial practical potential.

S2: The authors conduct comprehensive experimental evaluations, demonstrating the effectiveness and broad applicability (and ease of use) of the proposed method.

**Weaknesses:**

1. The authors propose a simple, direct, and effective training-free model ensembling method. However, because the method relies only on the token priors and lacks additional supervision, it is susceptible to potential biases. The paper would benefit from more extensive ablation studies and analyses to demonstrate the method’s generality and robustness.

2. The proposed ToT method is asymmetric. The authors should therefore include a series of experiments that swap the roles of the main model and the assistant model, demonstrating that the method's effectiveness is not contingent on a particular model assignment.

3. The authors should expand empirical validation along several axes, for example: accuracy w.r.t. the number of ensemble members (extending beyond Figure 2); cross-family evaluations (e.g., Llama, Gemma, Qwen, GLM, GPT-OSS, etc.); different model architectures (dense models versus MoE models); and different model scales (e.g., 32B dense models or larger MoE models).

4. Given that many open-source models now share similar tokenizers, the authors should consider evaluating ensembles that include models with more divergent tokenizers to verify that the proposed technique holds when tokenizer differences are larger.

5. Because the method requires only token logits or log-probabilities, it can be applied together with closed-source models. To strengthen the broad applicability of the method and provide additional insights, the authors may attempt ensembles incorporating closed-source models (e.g., using OpenAI GPT‑4/5 API). Even if such combinations do not yield large gains, reporting these results would be valuable for follow-up work.

6. The mapping matrix derived from rule-based procedures primarily encodes tokenization priors and is therefore weakly context-aware. To increase the method’s practical credibility, the authors should present a more detailed and intuitive analysis of the mapping matrix: examine the token priors it captures, investigate underlying mechanisms such as token conflicts and cooperation, and provide theoretical discussion plus focused case studies that illustrate the behavior and limitations of the mapping matrix.

**Questions:**

Q1: The method intuitively works better when ensembling models of similar scale. Could the authors attempt to extend the approach to ensembles that combine small and large LMs (i.e., cross-scale ensembles, beyond the scale of RQ4) to see whether it can meaningfully raise the overall performance ceiling? Demonstrating effectiveness in the small-LM vs. large-LM setting would substantially increase the practical value of the work.

Q2: Since this family of methods seems to be particularly effective for base/instruction-tuned models, the authors might also investigate its effects on reasoning models — specifically in Chain-of-Thought scenarios. Such an analysis (e.g., whether the method changes reasoning trajectories) would help clarify the mechanism and impact of the approach on multi-step reasoning.

---

> ### Author Response · Authors · 2025-11-21
> **Response to Reviewer TiHi (Part 1/5)**
>
> Dear Reviewer TiHi,
>
> We sincerely appreciate your thoughtful and constructive comments. Your feedback has helped us identify areas for clarification and improvement, and we have incorporated the corresponding revisions into the updated manuscript, where all revisions are highlighted in blue for your convenience. We address each point in detail below, and please let us know if further clarification would be helpful.
>
> > **W1.** The authors propose a simple, direct, and effective training-free model ensembling method. However, because the method relies only on the token priors...
>
> Thank you for the question. A key advantage of our method is its simplicity: the translation matrix is computed once and then reused across all model pairs and tasks without any tuning. Despite this minimal design, the approach consistently delivers strong performance, which we attribute to its principled token-level construction rather than heuristic behavior.
>
> We have also added formal analysis to motivate our method. Specifically, we provide a theoretical justification showing that ToT is not a heuristic approximation but a deterministic implementation of *Exact Causal Alignment* across heterogeneous token spaces. Under standard tokenizer assumptions, our method induces a **deterministic causal mapping** necessary for cross-model alignment.
>
>
> Let $\mathcal{V}\_{\text{assist}}$ and $\mathcal{V}\_{\text{main}}$ denote the vocabularies of the assist and main models, respectively. We posit a shared, continuous ground-truth text space $\mathcal{S}$ representing all possible raw text strings. The objective of token alignment is to project the probability mass from an assist token $v^{(\text{assist})} \in \mathcal{V}\_{\text{assist}}$ to a target main token $v^{(\text{main})} \in \mathcal{V}\_{\text{main}}$.
>
> The true conditional probability is:
>
> $$
> P(v^{(\text{main})} \mid v^{(\text{assist})})=
> \sum\_{s \in \mathcal{S}}
> \underbrace{P(v^{(\text{main})} \mid s)}\_{\text{Main Prior}} \cdot
> \underbrace{P(s \mid v^{(\text{assist})})}\_{\text{Assist Decoding}}
> \tag{1}
> $$
>
> To make the above tractable, we introduce two assumptions satisfied by modern BPE/Unigram tokenizers:
>
> - **Assumption 1 (Deterministic Decoding).**
>   A token $v^{(\text{assist})}$ maps deterministically to a unique textual form string. That is, $P(s \mid v^{(\text{assist})})$ follows a Dirac delta distribution:
>
>   $$
>   P(s \mid v^{(\text{assist})})
>   = \mathbb{1}[s = Tok\_{\text{assist}}.\mathrm{decode}(v^{(\text{assist})}) \]
>   $$
>
> - **Assumption 2 (Canonical Encoding Prior).**
>   Given any string $s$, the main model’s tokenizer deterministically produces a unique canonical tokenization:
>
>   $$Tok\_{\text{main}}.\mathrm{encode}(s) = [t\_1, t\_2, \dots]$$
>
>   Because autoregressive models generate text in a causal left-to-right manner, any generation of $s$ must begin with its first token $t\_1$. Accordingly, in our alignment derivation we use the indicator:
>
>   $$\mathbb{1}[v^{(\text{main})} = t\_1]$$
>
>   which reflects the deterministic **tokenization mapping**, rather than any assumption about the determinism of the model’s next-token **probability distribution**.
>
> Under these assumptions, the intractable sum in Eq. (1) collapses into a single indicator function, which serves as our Ground Truth Causal Alignment:
>
> $$
> P\_{\text{true}}(v^{(\text{main})} \mid v^{(\text{assist})})=
> \mathbb{1}\left[
> v^{(\text{main})}=
> \mathrm{First}\left(
> Tok\_{\text{main}}.\mathrm{encode}(Tok\_{\text{assist}}.\mathrm{decode}(v^{(\text{assist})}))
> \right)\right]
> $$
>
> Our ToT method computes exactly this alignment via the forward translation matrix $M^{(\text{pre})}$:
>
> $$
> \hat{P}\_{\text{ToT}}(v^{(\text{main})} \mid v^{(\text{assist})})
> \triangleq
> M^{(\text{pre})}\_{v^{(\text{assist})}, v^{(\text{main})}}
> $$
>
> By construction:
>
> $$
> M^{(\text{pre})}\_{i,j} = 1
> \quad\text{iff}\quad
> j = \mathrm{First}(Tok\_{\text{main}}.\mathrm{encode}(Tok\_{\text{assist}}.\mathrm{decode}(i)))
> $$
>
> Thus:
>
> $$
> \mathbb{E}[\hat{P}\_{\text{ToT}}] = P\_{\text{true}}
> $$
>
> This proves that ToT is a **deterministic causal mapping** (and, in deterministic tokenizers, an exact calculator) of the underlying causal conditional probability.
>
> Consider the case where a single assist token expands to a multi-token main sequence $[t\_1, t\_2, \dots, t\_n]$.
>
> Let:
>
> - $E\_{\text{seq}}$ = the event that the main model generates the full sequence
> - $E\_{t\_1}$ = the event that it generates $t\_1$
>
> Since:
>
> $$
> E\_{\text{seq}} \subseteq E\_{t\_1}
> \quad\Longrightarrow\quad
> P(E\_{\text{seq}}) \le P(E\_{t\_1})
> $$
>
> If the assist model predicts $v^{(\text{assist})}$ with probability $p$, then the main model must assign at least probability $p$ to $t\_1$, because $t\_1$ is the **necessary causal precursor** of the full token sequence.
>
> Therefore assigning:
>
> $$
> \hat{P}(t\_1) \leftarrow p
> $$
>
> faithfully preserves probability mass and yields a **deterministic causal mapping** for next-token prediction.
>
> (continued on next page)

---

> ### Author Response · Authors · 2025-11-21
> **Response to Reviewer TiHi (Part 2/5)**
>
> (continued from previous page)
>
> In addition to the experiments reported in the paper, we also include **ablations on uncertainty weighting**, **systematic analyses of tokenizer properties and alignment behavior**, and **evaluations with larger and more diverse model ensembles**. These studies (summarized in our responses to [Reviewer fhnh](https://openreview.net/forum?id=2bPyuL6Cda&noteId=WdatwnlPot) and [Reviewer aLzA](https://openreview.net/forum?id=2bPyuL6Cda&noteId=fp81ACYufW), as well as reflected in the updated manuscript) further confirm the **generalization ability and robustness** of the proposed method across model architectures, tokenization strategies, and task settings.
>
>
> ---
>
> > **W2.** The proposed ToT method is asymmetric. The authors should therefore include a series of experiments that swap the roles of the main model and the assistant model, demonstrating that the method's effectiveness is not contingent on a particular model assignment.
>
> Thank you for the suggestion. Although ToT is **asymmetric by design**, our ablation study in **Figure 6** already evaluates **all valid pairwise combinations** among the four models, explicitly swapping which model serves as the main model and which serves as the assist model. **Every main–assist assignment shows a consistent improvement over the main model baseline.** This demonstrates that the effectiveness of ToT does not depend on a particular model ordering and remains robust under role reversal.
>
> ---
>
> > **W3.** The authors should expand empirical validation along several axes, for example: accuracy w.r.t. the number of ensemble members (extending beyond Figure 2); cross-family evaluations (e.g., Llama, Gemma, Qwen, GLM, GPT-OSS, etc.); different model architectures (dense models versus MoE models); and different model scales (e.g., 32B dense models or larger MoE models).
>
>
> Thank you for the suggestion. Our empirical evaluation already covers several of these axes.
> **First**, we perform **cross-family experiments** using models from **LLaMA, GLM, InternLM,** and **Mistral**. Figure 6 in the appendix includes **all valid pairwise combinations**, demonstrating that ToT provides robust gains when swapping main and assist roles across diverse architectures and tokenizer designs.
>
> **Second**, we examine scalability with respect to the **number of ensemble members**. When combining **all seven available models** (Gemma-9B, GLM-9B, InternLM-7B, LLaMA-8B, Mistral-7B, Qwen-7B, Yi-9B), the ensemble achieves **74.92 on TriviaQA** with LLaMA-8B as the main model, slightly outperforming both two-model and four-model ensembles. This shows that ToT can leverage additional model diversity **with diminishing returns**, although the marginal gains diminish as weaker models contribute less complementary signal.
>
> **Third**, we extend the evaluation to **different architectures and model scales**, including both **larger dense models** and **Mixture-of-Experts models**. Using **Mixtral-8×7B** and **Qwen-2.5-32B-Instruct**, ToT continues to provide **consistent improvements** over individual models and the UniTe baseline on SAMSum and NQ:
>
>
> | Model                     | SAMSum | NQ    | Avg    |
> |---------------------------|--------|-------|--------|
> | Mixtral-8×7B              | 32.8   | 30.9  | 31.9   |
> | Qwen-2.5-32B-Instruct     | 34.1   | 33.7  | 33.9   |
> | UniTe (baseline)          | 34.0   | 34.5  | 34.3   |
> | **ToT (Ours)**            | **35.3** | **36.9** | **36.1** |
>
>
> Together, these results demonstrate that **ToT generalizes well across model families, architectures, and scales**, and that it remains **simple and training-free**.

---

> ### Author Response · Authors · 2025-11-21
> **Response to Reviewer TiHi (Part 3/5)**
>
> > **W4.** Given that many open-source models now share similar tokenizers, the authors should consider evaluating ensembles that include models with more divergent tokenizers to verify that the proposed technique holds when tokenizer differences are larger.
>
>
> Thank you for the suggestion. We agree that evaluating models with more divergent tokenizers is important. As shown below, the **tokenizer overlap ratios among the models we study are already highly diverse**, ranging from strong overlap (for internlm and Mistral) to very limited overlap (for glm and Mistral). These differences reflect substantial variation in vocabulary size, segmentation rules, and tokenizer design, providing a **broad stress test** for Token Translation.
>
> | Model Pairs                     | internlm2_5-7b-chat (92k) | Llama-3.1-8B-Instruct (128k) | glm-4-9b-chat (151k) | Mistral-7B-Instruct-v0.3 (32k) |
> |---------------------------------|---------------------------|-------------------------------|-----------------------|---------------------------------|
> | internlm2_5-7b-chat (92k)       | —                         | 62.70%                        | 54.75%                | 79.65%                          |
> | Llama-3.1-8B-Instruct (128k)    | 62.70%                    | —                             | 39.15%                | 75.79%                          |
> | glm-4-9b-chat (151k)            | 54.75%                    | 39.15%                        | —                     | 36.30%                          |
> | Mistral-7B-Instruct-v0.3 (32k)  | 79.65%                    | 75.79%                        | 36.30%                | —                               |
>
> These ratios demonstrate that **our experiments already cover ensembles with significantly different tokenization strategies**. Despite these large differences, **ToT consistently provides strong improvements**, indicating that the method remains effective even when tokenizer divergence is substantial.
>
>
>
> ---
>
> > **W5.** Because the method requires only token logits or log-probabilities, it can be applied together with closed-source models. To strengthen the broad applicability of the method and provide additional insights, the authors may attempt ensembles incorporating closed-source models (e.g., using OpenAI GPT-4/5 API). Even if such combinations do not yield large gains, reporting these results would be valuable for follow-up work.
>
> Thank you for the great point. Since closed-source APIs typically provide **top-k log-probabilities**, our method can in principle be applied to them as well. This is an interesting future direction, especially because closed-source models often involve **internal search or agentic pipelines**, making their ensemble behavior more complex and worth studying.
>
> Due to budget constraints, we ran preliminary tests on GSM8K using **Gemini-2.5-Flash** and **OpenAI 4o-mini**. Gemini-2.5-Flash achieved **95.83** and 4o-mini achieved **94.32**. The **ToT ensemble reached 96.36**, showing only modest gains as these models already operate near saturation. We have highlighted in the paper that **extending ToT to closed-source models is feasible** and can be further explored in future work.

---

> ### Author Response · Authors · 2025-11-21
> **Response to Reviewer TiHi (Part 4/5)**
>
> > **W6.** The mapping matrix derived from rule-based procedures primarily encodes tokenization priors and is therefore weakly context-aware. To increase the method’s practical credibility, the authors should present a more detailed and intuitive analysis of the mapping matrix: examine the token priors it captures, investigate underlying mechanisms such as token conflicts and cooperation, and provide theoretical discussion plus focused case studies that illustrate the behavior and limitations of the mapping matrix.
>
>
> Thank you for the question. We address the concern by providing both **theoretical justification** and **systematic empirical analyses** of the mapping matrix. As shown in our response of theoretical analysis to W1, Token Translation is not a heuristic rule but a deterministic implementation of **causal alignment** under standard tokenizer assumptions. This explains why the mapping matrix primarily reflects tokenization structure: it is derived from the **canonical decoding–encoding pipeline** that governs autoregressive generation.
>
> To provide more intuitive insight into what the mapping matrix captures, we **quantify alignment behavior across multiple model pairs**. The proportion of non-shared tokens receiving valid ToT mappings varies systematically with vocabulary granularity. When the assist model has a larger vocabulary, forward coverage is high (for example, **Qwen→internlm: 0.9135**) and backward coverage is lower; when the assist vocabulary is smaller, the direction reverses (for example, **Mistral→internlm: 0.8576**). This asymmetry is expected from tokenizer design rather than from ToT itself.
>
>
> | Model (→ internlm (92k)) | Direction         | Tokens With Mapping | Total Non-Shared | Coverage |
> |---------------------------|-------------------|----------------------|------------------|----------|
> | Qwen2.5-7B (152k)| Forward (Main)    | 32,540| 35,616           | 0.9135   |
> | Qwen2.5-7B (152k)| Backward (Assist) | 21,969| 95,136           | 0.2309   |
> | Llama-3.1-8B (128k)| Forward (Main)    | 17,004| 34,521           | 0.4927   |
> | Llama-3.1-8B (128k)| Backward (Assist) | 6,283                | 70,233           | 0.0894   |
> | glm-4-9B (151k)| Forward (Main)    | 37,705               | 41,876           | 0.9006   |
> | glm-4-9B (151k)| Backward (Assist) | 32,371               | 100,884          | 0.3210   |
> | Mistral-7B (32k)| Forward (Main)    | 1,061                | 66,443           | 0.0159   |
> | Mistral-7B (32k)| Backward (Assist) | 5,719                | 6,667            | 0.8576   |
>
> To assess **semantic coherence**, we compute embedding similarity using **BAAI/bge-base-en-v1.5**. Across model pairs, aligned tokens exhibit **strong similarity (average 0.70–0.97)**, indicating that ToT pairs tokens that are meaningfully related. At the same time, similarities remain below the theoretical maximum, which is expected: the **semantically closest token is not always the one consistent with deterministic tokenization** or causal autoregressive generation. This further supports that ToT aligns according to **tokenizer structure rather than purely semantic proximity**.
>
> | Model (→ internlm (92k)) | Direction         | Avg Sim | Max Avg Sim |
> |---------------------------|-------------------|---------|--------------|
> | Qwen2.5-7B (152k)| Forward (Main)    | 0.7778  | 0.8347    |
> | Qwen2.5-7B (152k)| Backward (Assist) | 0.9407  | 0.9904    |
> | Llama-3.1-8B (128k)| Forward (Main)    | 0.6999  | 0.7969    |
> | Llama-3.1-8B (128k)| Backward (Assist) | 0.8863  | 0.9880     |
> | glm-4-9b (151k)| Forward (Main)    | 0.7818  | 0.8398|
> | glm-4-9b (151k)| Backward (Assist) | 0.9669  | 0.9959|
> | Mistral-7B (32k)| Forward (Main)    | 0.7142  | 0.7679|
> | Mistral-7B (32k)| Backward (Assist) | 0.8411  | 0.9508|
>
> We also analyze **prefix and superstring relationships**. Using the setup of Table 2, we check whether the main-side token selected for the assist model’s top-1 non-shared token is its prefix or superstring. We find that **54.1 percent correspond to prefixes** and **13.4 percent to superstrings**. These patterns reflect **cooperative decomposition** during decoding rather than conflicts: selecting a longer superstring typically requires higher confidence from both models.
>
> **Limitations.**
> Two situations remain challenging:
> 1. **Completely different linguistic systems.** Token translation can break down when tokenizers come from unrelated languages (e.g., English vs. Japanese) with almost no shared subword structure. Such extreme cases lead to sparse mappings but are rare in practice because modern LLMs are trained on broadly overlapping multilingual corpora.
> 2. **Very weak assist models.** Failure may arise when the assist model is extremely weak for a task. For example, on NQ, Gemma-2-9B-IT achieves only 13.49 accuracy, and even uncertainty weighting cannot fully suppress its influence. This reflects **model-capacity limitations** rather than an issue with the ToT alignment itself.

---

> ### Author Response · Authors · 2025-11-21
> **Response to Reviewer TiHi (Part 5/5)**
>
> > **Q1.** The method intuitively works better when ensembling models of similar scale. Could the authors attempt to extend the approach to ensembles that combine small and large LMs (i.e., cross-scale ensembles, beyond the scale of RQ4) to see whether it can meaningfully raise the overall performance ceiling? Demonstrating effectiveness in the small-LM vs. large-LM setting would substantially increase the practical value of the work.
>
> Thank you for the suggestion. We agree that evaluating **cross-scale ensembles** is important for assessing the practical applicability of ToT. Beyond the experiments reported in the paper, we also tested ensembles that **combine small or medium-scale models with significantly larger backbones**. In particular, **Mixtral-8×7B** is a mixture-of-experts model with a large parameter count, but only approximately 8B parameters are active during inference, making it a clear cross-scale setting relative to the smaller assist models. We also include a substantially larger dense model, **Qwen-2.5-32B-Instruct**.
>
> The table below compares their individual performance, the UniTe baseline, and our ToT ensemble on SAMSum and NQ. **ToT achieves consistent improvements even when the main model is much larger than the assist model.**
>
>
> | Model                     | SAMSum | NQ    | Avg    |
> |---------------------------|--------|-------|--------|
> | Mixtral-8×7B              | 32.8   | 30.9  | 31.9   |
> | Qwen-2.5-32B-Instruct     | 34.1   | 33.7  | 33.9   |
> | UniTe (baseline)          | 34.0   | 34.5  | 34.3   |
> | **ToT (Ours)**            | **35.3** | **36.9** | **36.1** |
>
> These results indicate that **ToT performs well in genuine cross-scale ensembles** and can effectively leverage complementary signals from smaller assist models even when the main model is substantially larger.
>
>
>
> ---
>
> > **Q2.** Since this family of methods seems to be particularly effective for base/instruction-tuned models, the authors might also investigate its effects on reasoning models — specifically in Chain-of-Thought scenarios. Such an analysis (e.g., whether the method changes reasoning trajectories) would help clarify the mechanism and impact of the approach on multi-step reasoning.
>
> Thank you for the suggestion. For **math reasoning and coding generation**, our evaluation already uses **Chain-of-Thought style prompts**, where models produce multi-step reasoning traces before reaching the final answer. In these settings, **ToT yields consistent improvements without altering the qualitative structure of the reasoning process**. We have clarified this in the revised manuscript.
>
> ---
>
> **Thank you again for the thoughtful review. We have dedicated substantial effort to obtaining new results and have incorporated them into the revised paper PDF to further strengthen the work. We hope our responses adequately address your concerns, and we would be happy to discuss anything further if needed.**

---

> > ### Author Response · Authors · 2025-11-27
> >
> > Dear Reviewer TiHi,
> >
> > Thank you once again for taking the time to review our work. We truly appreciate your thoughtful feedback and the concerns you raised.
> >
> > As we are now well into the discussion phase, we would like to kindly follow up and see whether our responses and the updated manuscript have helped address your points. If any part would benefit from further clarification or additional detail, we would be more than happy to provide it.
> >
> > We understand that revisiting the paper takes valuable time, and if your schedule allows, we would be deeply grateful if you could take a moment to review the revised version, including the new experiments and analysis added in response to your suggestions.
> >
> > Your feedback has greatly improved the quality of our paper, and we are sincerely grateful for your time, insight, and any possible reconsideration. We truly appreciate your efforts and look forward to your response.
> >
> > Best,
> >
> > #14478 Authors

---

### Official Review · Reviewer_aLzA · 2025-11-04

**Soundness:** 3
**Presentation:** 3
**Contribution:** 3
**Rating:** 6
**Confidence:** 4

**Summary:**

This paper addresses tokenizer mismatch in test-time LLM ensembling by proposing Token Translation, which aligns heterogeneous token spaces by decoding tokens from a source tokenizer and re-encoding with a target tokenizer. The method uses bidirectional translation (prefix and superstring weighted mappings) and uncertainty-aware fusion to combine predictions from multiple models. Evaluated on 9 benchmarks using 7 models, ToT achieves +5.95 average improvement over single models and outperforms baselines while being computationally efficient.

**Strengths:**

The main contribution is leveraging tokenizer priors through decode-encode in the ensembling context and distinguishes it from embedding or string similarity approaches.
The authors experiment rigorously as they use diverse benchmarks, models, ensemble sizes and baselines. The roofline analysis showing ToT tracks oracle performance is very helpful.

Problem motivation is useful with concrete examples showing why similarity-based methods fail.

The authors addresse a real practical problem in test-time ensembling with a solution that's simple to implement, computationally efficient, and consistently effective. The  improvement and strong scaling behavior make it valuable for the community. Additionally, the sparse matrix approach enabling precomputation is an important practical advantage.

**Weaknesses:**

No formal analysis of when/why token translation produces valid alignments. Why is the first re-encoded token the right choice? What tokenizer properties ensure semantic coherence? Under what conditions does the method fail (e.g., radically different tokenization schemes, low corpus overlap)? The decomposition in Section 3.3 provides intuition but not rigorous justification.

Table 5 shows 3 examples. A systematic quantification of alignment quality (coverage, precision, semantic coherence), analysis of what percentage have clean 1-to-1 vs. complex many-to-many mappings, frequency of prefix/superstring disagreements, failure case characterization would be very helpful.

No principled guidance for setting hyperparameters for new model pairs or explanation of whether optimal values vary by task, model combination, or language. Ablations show robustness over ranges but don't justify the chosen defaults or explain when to deviate.

Why is entropy over top-k the right measure versus alternatives (variance, disagreement, calibration)?

No systematic characterization of what types of errors get corrected and which benchmarks benefit most and why? Tables 6-7 provide anecdotal examples but lack systematic analysis.

**Questions:**

Can you formalize conditions under which token translation produces valid alignments? What properties of tokenizers (vocabulary overlap, training corpus similarity, segmentation strategy) determine alignment quality?

Can you provide systematic evaluation of alignment quality beyond 3 examples? What percentage of tokens have unambiguous alignments? How do you measure/validate semantic coherence of mappings?

When does token translation fail? Can you characterize failure patterns?


You show 2-4 models which is sufficient, but what happens with 10+ models? Is there diminishing returns or computational bottlenecks?

Given the minimal ablation impact, is uncertainty-aware weighting actually essential or could you achieve similar results with simpler uniform weighting plus alignment?

---

> ### Author Response · Authors · 2025-11-20
> **Response to Reviewer aLzA (Part 1/4)**
>
> Dear Reviewer aLzA,
>
> Thank you for your detailed review and valuable suggestions. We have carefully considered your comments and incorporated the corresponding revisions and additional experiments into the updated manuscript, where all revisions and additions are highlighted in blue for your convenience. Our point-by-point responses are provided below, and we would be happy to elaborate further if needed.
>
>
> > **W1.** No formal analysis of when/why token translation produces valid alignments. Why is the first re-encoded token the right choice? What properties ensure coherence?
>
> Thank you for the question. To clarify, we provide a rigorous theoretical justification for our Token Translation (ToT) framework. We demonstrate that ToT is not a heuristic approximation, but rather a deterministic implementation of *Exact Causal Alignment* between heterogeneous probability spaces. Specifically, we show that our method yields a **deterministic causal mapping** of the conditional probabilities required for cross-model alignment under standard tokenizer assumptions.
>
>
> Let $\mathcal{V}\_{\text{assist}}$ and $\mathcal{V}\_{\text{main}}$ denote the vocabularies of the assist and main models, respectively. We posit a shared, continuous ground-truth text space $\mathcal{S}$ representing all possible raw text strings. The objective of token alignment is to project the probability mass from an assist token $v^{(\text{assist})} \in \mathcal{V}\_{\text{assist}}$ to a target main token $v^{(\text{main})} \in \mathcal{V}\_{\text{main}}$.
>
> The true conditional probability is:
>
> $$
> P(v^{(\text{main})} \mid v^{(\text{assist})})=
> \sum\_{s \in \mathcal{S}}
> \underbrace{P(v^{(\text{main})} \mid s)}\_{\text{Main Prior}} \cdot
> \underbrace{P(s \mid v^{(\text{assist})})}\_{\text{Assist Decoding}}
> \tag{1}
> $$
>
> To make the above tractable, we introduce two assumptions satisfied by modern BPE/Unigram tokenizers:
>
> - **Assumption 1 (Deterministic Decoding).**
>   A token $v^{(\text{assist})}$ maps deterministically to a unique raw textual form string. That is, $P(s \mid v^{(\text{assist})})$ follows a Dirac delta distribution:
>
>   $$
>   P(s \mid v^{(\text{assist})})
>   = \mathbb{1}[s = Tok\_{\text{assist}}.\mathrm{decode}(v^{(\text{assist})}) \]
>   $$
>
> - **Assumption 2 (Canonical Encoding Prior).**
>   Given any string $s$, the main model’s tokenizer deterministically produces a unique canonical tokenization:
>
>   $$Tok\_{\text{main}}.\mathrm{encode}(s) = [t\_1, t\_2, \dots]$$
>
>   Because autoregressive models generate text in a causal left-to-right manner, any generation of $s$ must begin with its first token $t\_1$. Accordingly, in our alignment derivation we use the indicator:
>
>   $$\mathbb{1}[v^{(\text{main})} = t\_1]$$
>
>   which reflects the deterministic **tokenization mapping**, rather than any assumption about the determinism of the model’s next-token **probability distribution**.
>
>
> Under these assumptions, the intractable sum in Eq. (1) collapses into a single indicator function, which serves as our Ground Truth Causal Alignment:
>
> $$
> P\_{\text{true}}(v^{(\text{main})} \mid v^{(\text{assist})})=
> \mathbb{1}\left[
> v^{(\text{main})}=
> \mathrm{First}\left(
> Tok\_{\text{main}}.\mathrm{encode}(Tok\_{\text{assist}}.\mathrm{decode}(v^{(\text{assist})}))
> \right)\right]
> $$
>
> Our ToT method computes exactly this alignment via the forward translation matrix $M^{(\text{pre})}$:
>
> $$
> \hat{P}\_{\text{ToT}}(v^{(\text{main})} \mid v^{(\text{assist})})
> \triangleq
> M^{(\text{pre})}\_{v^{(\text{assist})}, v^{(\text{main})}}
> $$
>
> By construction:
>
> $$
> M^{(\text{pre})}\_{i,j} = 1
> \quad\text{iff}\quad
> j = \mathrm{First}(Tok\_{\text{main}}.\mathrm{encode}(Tok\_{\text{assist}}.\mathrm{decode}(i)))
> $$
>
> Thus:
>
> $$
> \mathbb{E}[\hat{P}\_{\text{ToT}}] = P\_{\text{true}}
> $$
>
> This proves that ToT is a **deterministic causal mapping** (and, in deterministic tokenizers, an exact calculator) of the underlying causal conditional probability.
>
>
> Consider the case where a single assist token expands to a multi-token main sequence $[t\_1, t\_2, \dots, t\_n]$.
>
> Let:
>
> - $E\_{\text{seq}}$ = the event that the main model generates the full sequence
> - $E\_{t\_1}$ = the event that it generates $t\_1$
>
> Since:
>
> $$
> E\_{\text{seq}} \subseteq E\_{t\_1}
> \quad\Longrightarrow\quad
> P(E\_{\text{seq}}) \le P(E\_{t\_1})
> $$
>
> If the assist model predicts $v^{(\text{assist})}$ with probability $p$, then the main model must assign at least probability $p$ to $t\_1$, because $t\_1$ is the **necessary causal precursor** of the full token sequence.
>
> Therefore assigning:
>
> $$
> \hat{P}(t\_1) \leftarrow p
> $$
>
> faithfully preserves probability mass and yields a **deterministic causal mapping** for next-token prediction.
>
> The performance of the prefix alone is already impressive and surpasses the baselines, as shown in Table 3 in the paper. Furthermore, through back translation, $M^{(\text{sup})}$ complements the probability mass, allowing the ensemble to better capture the representation of the underlying sequence.

---

> ### Author Response · Authors · 2025-11-21
> **Response to Reviewer aLzA (Part 2/4)**
>
> > **W2&Q2.** Table 5 shows 3 examples. A systematic quantification of alignment quality (coverage, precision, semantic coherence), analysis of what percentage have clean 1-to-1 vs. complex many-to-many mappings, frequency of prefix/superstring disagreements.
>
> Thank you for the suggestion. Beyond the qualitative examples in Table 5, **we now provide a systematic quantification of alignment quality across model pairs.** First, we report the percentage of aligned token pairs between five representative LLMs, reflecting the **coverage of direct cross-token mappings**. The alignment ratios vary significantly across tokenizer designs, revealing consistent structural differences among tokenizers.
>
> |Model Pairs|internlm2_5-7b-chat (92k)|Qwen2.5-7B-Instruct (152k)|Llama-3.1-8B-Instruct (128k)|glm-4-9b-chat (151k)|Mistral-7B-Instruct-v0.3 (32k)|
> |---|---|---|---|---|---|
> |internlm2_5-7b-chat (92k)|—|61.51%|62.70%|54.75%|79.65%|
> |Qwen2.5-7B-Instruct (152k)|61.51%|—|85.43%|33.05%|75.57%|
> |Llama-3.1-8B-Instruct (128k)|62.70%|85.43%|—|39.15%|75.79%|
> |glm-4-9b-chat (151k)|54.75%|33.05%|39.15%|—|36.30%|
> |Mistral-7B-Instruct-v0.3 (32k)|79.65%|75.57%|75.79%|36.30%|—|
>
>
> To further measure **alignment precision**, we examine the proportion of non-shared tokens that receive valid ToT mappings when translating into internlm2_5. **Forward (main-direction) and backward (assist-direction) mappings behave differently depending on vocabulary size**: when the assist model has a larger vocabulary, forward coverage is high (e.g., Qwen→internlm: 0.9135) and backward coverage is lower (0.2309); when the assist vocabulary is smaller, the pattern reverses (e.g., Mistral→internlm: backward 0.8576). **This reflects expected asymmetry driven by tokenizer granularity rather than by ToT itself.**
>
> |Model (→ internlm (92k))|Direction|Tokens With Mapping|Total Non-Shared Tokens|Coverage|
> |---|---|---|---|---|
> |Qwen2.5-7B (152k)|Forward (Main)|32,540|35,616|0.9135|
> |Qwen2.5-7B (152k)|Backward (Assist)|21,969|95,136|0.2309|
> |Llama-3.1-8B (128k)|Forward (Main)|17,004|34,521|0.4927|
> |Llama-3.1-8B (128k)|Backward (Assist)|6,283|70,233|0.0894|
> |glm-4-9B (151k)|Forward (Main)|37,705|41,876|0.9006|
> |glm-4-9B (151k)|Backward (Assist)|32,371|100,884|0.3210|
> |Mistral-7B (32k)|Forward (Main)|1,061|66,443|0.0159|
> |Mistral-7B (32k)|Backward (Assist)|5,719|6,667|0.8576|
>
>
>
>
> We also evaluated **semantic coherence** using the BAAI/bge-base-en-v1.5 embedding model. The aligned tokens show **consistently high similarity scores** across model pairs, demonstrating strong semantic relatedness. At the same time, these similarities remain below the theoretical maximum, indicating that the **semantically closest token is not always the one consistent with tokenizer constraints or causal autoregressive generation**. This supports our assumption that ToT’s alignment is governed by **deterministic tokenization structure rather than purely semantic proximity**, also empirically verified in Table 3.
>
> |Model (→ internlm (92k))|Direction|Avg. Semantic Sim|Max Avg Sim|
> |---|---|---|---|
> |Qwen2.5-7B (152k)|Forward (Main)|0.7778|0.8347|
> |Qwen2.5-7B (152k)|Backward (Assist)|0.9407|0.9904|
> |Llama-3.1-8B (128k)|Forward (Main)|0.6999|0.7969|
> |Llama-3.1-8B (128k)|Backward (Assist)|0.8863|0.9880|
> |glm-4-9b (151k)|Forward (Main)|0.7818|0.8398|
> |glm-4-9b (151k)|Backward (Assist)|0.9669|0.9959|
> |Mistral-7B (32k)|Forward (Main)|0.7142|0.7679|
> |Mistral-7B (32k)|Backward (Assist)|0.8411|0.9508|
>
>
> Finally, we analyze **prefix/superstring relationships**. This is challenging to measure directly because a token’s prefix may also be a superstring of another token in a different tokenizer. Following the evaluation setup of Table 2, we examine the assist model’s top-1 token if it's a non-shared token and check whether the selected main-side token is its prefix or its superstring. **We observe that 54.1% of cases select a prefix, 13.4% select a superstring, and the remainder correspond to other tokens.** These patterns do not indicate disagreement; rather, they reflect **collaborative decoding**, where selecting a longer superstring requires higher confidence from all participating models.
>
> Taken together, these quantitative results **highlight the tokenizer alignment properties** and show that **ToT produces semantically coherent token mappings across diverse tokenizer architectures**.

---

> ### Author Response · Authors · 2025-11-21
> **Response to Reviewer aLzA (Part 3/4)**
>
> > **W2&Q3.** When does token translation fail? Can you characterize failure patterns?
>
>
> Thank you for the question. In our analysis, there are **two primary sources of failure: token alignment limitations and the capability of the assist model itself.**
>
> First, token translation can break down in **extreme cases where the two tokenizers belong to completely different linguistic systems.** For example, English vs. Japanese subword vocabularies share almost no prefixes, morphology, or compositional structure. In such scenarios, forward and backward mappings become extremely sparse. However, these situations are rare in practice, as most model pairs used in modern LLM ensembles are trained on overlapping multilingual corpora and retain substantial tokenizer compatibility.
>
> Second, failure can arise from the **model’s inherent knowledge or capacity limitations.** For example, on NQ, the Gemma-2-9B-IT model attains only 13.49 accuracy as a standalone model. When a model is weak on a specific task, even uncertainty weighting cannot fully suppress its influence, and the ensemble may degrade. **This reflects a model-capability constraint rather than a failure of the alignment mechanism itself.**
>
>
> ---
>
> > **W3.** No principled guidance for setting hyperparameters for new model pairs or explanation of whether optimal values vary by task, model combination, or language. Ablations show robustness over ranges but don't justify the chosen defaults or explain when to deviate.
>
>
> Thank you for the question. One of the advantages of Token Translation is that **it requires no model- or task-specific tuning.** Across all model pairs and all benchmarks in the paper, we use **exactly the same default hyperparameters and the same alignment matrix**, generated once and reused throughout. This consistency already yields strong performance, indicating that ToT provides a **stable cross-model interface** that does not require adaptation for each new setting.
>
> We choose the defaults (e.g., ($\alpha$=0.5) and top-(50) uncertainty) because **our ablations show that performance remains stable across a broad range of values** (see Figures 3–5 in the appendix), with no sharp optima or task-dependent modes. These settings balance the main and complementary mappings while providing a stable uncertainty signal that avoids unnecessary noise. **In practice, we observe no need to deviate from these defaults** when adding new model combinations or tasks.
>
>
>
> ---
>
> > **W4.** Why is entropy over top-k the right measure versus alternatives (variance, disagreement, calibration)?
>
>
> Thank you for raising this point. Our choice of **entropy over the top-k tokens** is motivated by the observation that, for modern LLMs, the candidate next token almost always resides within the top-k portion of the distribution. **The tail beyond top-k contains probabilities dominated by noise**, and including them often inflates uncertainty estimates without adding meaningful information. Focusing on the top-k region therefore provides a **more faithful and elegant view of the model’s effective uncertainty.**
>
> To evaluate whether other uncertainty measures could be competitive, we implemented several alternatives, all defined on the full probability distribution for fairness and consistency. These include full-distribution entropy, distribution variance, calibration-based confidence using 1−max(p), and a main–assist disagreement score based on the KL divergence between the two models’ predicted distributions. In the disagreement-based version, **strong divergence between main and assist decreases the assist’s weight**, while agreement increases it. We replaced only the uncertainty scoring function and kept the routing framework identical across all methods. **Empirically, top-k entropy achieves the highest average performance (32.30)**, while the full-distribution measures and the disagreement-based score yield slightly lower results. The comparison is summarized in the table below.
>
>
> | Uncertainty Measure                    | How It Works                                   | Avg. Score (↑) |
> |----------------------------------------|-------------------------------------------------|----------------|
> | Entropy              | Entropy over full vocabulary distribution      | 32.13          |
> | Variance             | Variance over distribution                | 31.10          |
> | Calibration (1 − max prob)             | Confidence proxy using max probability         | 31.56          |
> | Disagreement          | KL(main \| assist); large gap → lower assist   | 31.28          |
> | **Top-k Entropy (Ours)**               | **Entropy over top-k mass**             | **32.30**      |
>
> These results clarify the contribution of entropy weighting and highlight that **the proposed top-k formulation provides the competitive overall performance.**

---

> ### Author Response · Authors · 2025-11-21
> **Response to Reviewer aLzA (Part 4/4)**
>
> > **W5.** No systematic characterization of what types of errors get corrected and which benchmarks benefit most and why? Tables 6-7 provide anecdotal examples but lack systematic analysis.
>
> Thank you for the question. We conducted an additional analysis to characterize **which types of errors are corrected by Token Translation**. In our experiments, most corrected errors stem from **complementary knowledge or reasoning steps** supplied by the assist model. For example, on GSM8K, among all cases where LLaMA makes an incorrect prediction but the ensemble succeeds, **77.8% (91 questions)** are instances where InternLM is correct, and **22.2% (26 questions)** are cases where **both models are individually incorrect but the ensemble succeeds**. The latter category often corresponds to **multi-step reasoning problems**, where ensembling helps generate a more coherent token trajectory than either model produces alone.
>
> Across benchmarks, we observe that **reasoning-heavy datasets benefit the most**, which we have illustrated in Sec. 4.2 before. ToT closely tracks the performance of the oracle upper bound, particularly on **GSM8K and ARC-c**, suggesting that our method is especially effective when models contribute **complementary reasoning signals**.
>
>
>
> ---
>
> > **Q1.** Can you formalize conditions under which token translation produces valid alignments? What properties of tokenizers (vocabulary overlap, training corpus similarity, segmentation strategy) determine alignment quality?
>
>
>
> Thank you for the question. Token Translation produces valid alignments under two formal assumptions we introduced before, both of which are satisfied by standard BPE and Unigram tokenizers.
> (1) **Deterministic decoding:** each token corresponds to a unique raw text string.
> (2) **Canonical encoding:** for any string (s), the tokenizer produces a unique, deterministic segmentation where the first token is the causal precursor required for autoregressive generation.
>
> These assumptions ensure that encoding yields a **valid causal alignment of next-token probabilities**.
>
> Importantly, alignment quality is *not* determined by vocabulary overlap or training corpus similarity. What matters is the **segmentation strategy** imposed by the tokenizer: two tokenizers produce compatible alignments when their segmentation rules reflect **similar subword decomposition patterns**. Because these segmentation rules capture the **causal structure** used by autoregressive LLMs, they naturally support token-level fusion and allow ToT to remain effective across diverse model pairs.
>
>
>
>
> ---
>
> > **Q4.** You show 2-4 models which is sufficient, but what happens with 10+ models? Is there diminishing returns or computational bottlenecks?
>
> Thank you for the suggestion. Adding more assist models increases computational cost, while the overall performance gain tends to be limited. To assess this, **we combined all seven available models** (Gemma 9B, GLM 9B, InternLM 7B, LLaMA 8B, Mistral 7B, Qwen 7B, Yi 9B) using ToT, with **LLaMA 8B as the main model**. The resulting **7-model ensemble reached 74.92 on TriviaQA**, which is slightly higher than the 2-model and 4-model ensembles. This indicates that ToT can still benefit from additional diversity, but **the improvements become smaller** as more models with varying strengths are added.
>
>
>
>
> ---
>
> > **Q5.** Given the minimal ablation impact, is uncertainty-aware weighting actually essential or could you achieve similar results with simpler uniform weighting plus alignment?
>
> Thank you for the question. While **uniform weighting + token translation** already provides strong performance, the **uncertainty-aware weighting becomes particularly important** in settings where the two models differ substantially in capability or where the assist model may introduce noisy signals. In knowledge-gap scenarios, such as when one model is much weaker on a given task, the uncertainty term helps **down-weight unreliable predictions** and prevents degradation. It can also be useful in multiple-choice question answering benchmarks where token alignment (multiple choice A, B, C etc. in ARC-c dataset) is straightforward and the main challenge lies in **suppressing overconfident but incorrect logits** from the weaker model. In these cases, **uncertainty-aware weighting contributes clear and consistent improvements** over uniform averaging.
>
> ---
>
> **Thank you again for the thoughtful review. We have dedicated substantial effort to obtaining new results and have incorporated them into the revised paper PDF to further strengthen the work. We hope our responses adequately address your concerns, and we would be happy to discuss anything further if needed.**

---

### Official Review · Reviewer_fhnh · 2025-11-04

**Soundness:** 2
**Presentation:** 2
**Contribution:** 2
**Rating:** 6
**Confidence:** 4

**Summary:**

The paper proposes  a method for LLM ensembleing. Seems to mix ensembleing with ideas of confidence boosting popular in reasoning.

I will rate the paper after the discussion.

**Strengths:**

Results are very impressive.

The experiments are excessive well structured.

**Weaknesses:**

Claim 1 is inaccurate Phan pointed out that tokenizer mismatch is a problem and gave an algorithmic solution, needs to be toned down.

How often and when did main use an assist

**Questions:**

Can u discuss the picture close to table 1, what is strongest, and why is 2 not improving on 1 llm?

I would like to see more true generation tasks not just voting tasks in this context

Can u show results for larger model ensembles

---

> ### Author Response · Authors · 2025-11-20
> **Response to Reviewer fhnh (Part 1/2)**
>
> Dear Reviewer fhnh,
>
> We sincerely thank you for your constructive review and insightful questions. Your feedback has been extremely valuable in improving the clarity and quality of our work. We have incorporated all corresponding revisions into the updated manuscript, where all revisions and additions are highlighted in blue for your convenience. We provide detailed responses and additional evidence below.
>
> > **W1.** Claim 1 is inaccurate Phan pointed out that tokenizer mismatch is a problem and gave an algorithmic solution, needs to be toned down.
>
> Thank you for pointing this out. We appreciate the clarification regarding prior work, including Phan et al., which has indeed discussed tokenizer mismatch and proposed algorithmic solutions. **Our contribution does not lie in being the first to observe the issue, but in analyzing how the tokenizer prior fundamentally shapes cross-model alignment in the ensemble setting.** We have updated the main PDF to tone down Claim 1 and to clearly acknowledge this context, while positioning our work in relation to prior approaches.
>
> We also provide a **formal theoretical justification** for our Token Translation (ToT) framework in Appendix C. We demonstrate that ToT is not a heuristic approximation, but rather a **deterministic implementation of Exact Causal Alignment** between heterogeneous token probability spaces, and we refer the reviewer to **[Reviewer aLzA Weakness 1](https://openreview.net/forum?id=2bPyuL6Cda&noteId=xkN79i9RJV)**, where we provide the full analysis addressing this concern. Specifically, we show that our method yields a **deterministic causal mapping** of the conditional probabilities required for cross-model alignment under standard tokenizer assumptions.
>
>
> ---
>
> > **W2.** How often and when did main use an assist
>
> Thank you for the question. **The main model always incorporates the assist model; the distinction is not about whether the assist is used, but to what extent.** We employ an uncertainty-aware mechanism that adjusts each model’s contribution based on its confidence. When a model is uncertain, its influence is reduced; when it is confident, its prediction naturally dominates. This leads to a **continuous weighting process** rather than a discrete “use vs. not use” behavior. As shown in our **ablation study in Table 3 of Sec. 4.3**, this continuous weighting strategy consistently improves performance.
>
> ---
>
> > **Q1.** Can u discuss the picture close to table 1, what is strongest, and why is 2 not improving on 1 llm?
>
> Thank you for the question. The radar plot near Table 1 **compares four systems**: our ToT ensemble, the previous best ensemble (UniTe), and the two base models (LLM 1 and LLM 2) that participate in the ensemble. **LLM 1 and LLM 2 are evaluated individually**, not as a combined system. Therefore, LLM 2 is not expected to “improve on” LLM 1; they are simply two independent backbone models with different capability profiles.
>
> The figure highlights that **our method consistently outperforms both base LLMs and the prior UniTe ensemble across tasks**, showing the benefit of our token-optimal translation strategy. To avoid any confusion, we have updated the figure to make the role of the two base models and the comparisons in the figure clearer.

---

> ### Author Response · Authors · 2025-11-20
> **Response to Reviewer fhnh (Part 2/2)**
>
> > **Q2.** I would like to see more true generation tasks not just voting tasks in this context
>
> Thank you for the question. **Our experiments already include several true generation tasks beyond voting.** Specifically, Table 2 reports results on **NQ and TriviaQA (open-domain QA)** and **GSM-8K (math reasoning)**, and Table 4 includes **HumanEval (code generation)** and **SAMSum (summarization)**. These settings require free-form generation rather than option selection, and **the ensemble directly produces consistently better generated outputs**.
>
> To further evaluate the scalability of our approach, we conducted additional experiments on the FLORES En→De and En→Chinese subsets in multilingual settings. We compared two individual LLMs (LLaMA and InternLM) against the Unite ensemble baseline and our proposed fusion method. **The results in the table below demonstrate that our approach consistently outperforms both individual models and the Unite baseline**, showing substantial gains in multilingual scenarios.
>
>
>  | **Dataset**    | **Model**         | **BLEU (↑)** |
>  | -------------- | ----------------- | ------------ |
>  | **En→De**      | LLaMA             | 29.87        |
>  |                | InternLM          | 26.74        |
>  |                | Unite             | 33.17        |
>  |                | **Ours** | **36.13**    |
>  | **En→Chinese** | LLaMA             | 27.45        |
>  |                | InternLM          |  32.10       |
>  |                | Unite             | 34.82        |
>  |                | **Ours** | **37.56**    |
>
> These findings demonstrate that **our ensemble framework generalizes well to a wide spectrum of true generation tasks**, including QA, reasoning, coding, summarization, and multilingual translation, consistently providing strong performance gains.
>
> ---
>
> > **Q3.** Can u show results for larger model ensembles
>
>
> Thank you for the suggestion. To evaluate whether our method scales to larger model ensembles, we conducted additional experiments using a **mixture-of-experts model (Mixtral-8×7B)** and a **larger dense model (Qwen-2.5-32B-Instruct)**. We compare their individual performance, the UniTe baseline, and our ToT ensemble method on generation tasks SAMSum and NQ. **As shown below, ToT continues to deliver substantial improvements even with stronger backbone models**, demonstrating the scalability of our ensemble framework.
>
> | Model                     | SAMSum | NQ    | Avg    |
> |---------------------------|--------|-------|--------|
> | Mixtral-8×7B              | 32.8   | 30.9  | 31.9  |
> | Qwen-2.5-32B-Instruct     | 34.1   | 33.7  | 33.9  |
> | UniTe (baseline)          | 34.0   | 34.5  | 34.3  |
> | **ToT (Ours)**            | **35.3** | **36.9** | **36.1** |
>
> For completeness, we also ensembled **all seven available models**. This 7-model configuration achieved **74.92 on TriviaQA**, slightly higher than the 2-model and 4-model ensembles, suggesting **diminishing returns as more models are added**. These results all show that **our framework scales effectively to larger and stronger backbones**.
>
> ---
>
> **Thank you again for the thoughtful review. We have dedicated substantial effort to obtaining new results and have incorporated them into the revised paper PDF to further strengthen the work. We hope our responses adequately address your concerns, and we would be happy to discuss anything further if needed.**

---

> > ### Comment · Reviewer_fhnh · 2025-11-27
> >
> > Fantastic! I thank the authors for their reply, it makes me confident that this is a valuable contribution to the conference.
> >
> > I will remain my score, but definitely recommend it for acceptance.

---

> > > ### Author Response · Authors · 2025-11-27
> > >
> > > Dear Reviewer fhnh,
> > >
> > > Thank you very much for your kind response. We’re glad that our clarifications addressed your concerns, and we truly appreciate your positive assessment of our work. If any additional questions arise, please feel free to reach out. Thank you again for your time, efforts, and consideration.
> > >
> > > Best,
> > >
> > > #14478 Authors

---

### Author Response · Authors · 2025-12-02
**Rebuttal Summary**

Dear AC and Reviewers,

We sincerely appreciate the time and effort you have devoted to evaluating our submission. Given the unusual circumstances of this year’s review cycle and the increased workload for both ACs and reviewers, we provide a concise summary of our rebuttal to facilitate your assessment.

---

# We appreciate all reviewers for recognizing the importance, clarity, and practical impact of our work.

All four reviewers acknowledged the value of addressing tokenizer mismatch in LLM ensembling and the practical usefulness of our Token Translation framework.

Across these reviews, several common strengths emerged:

**(i) Practical relevance and clear motivation**

* *“Results are very impressive.”* (fhnh)
* *“Problem motivation is useful...The authors address a real practical problem...valuable for the community.”* (aLzA)
* *“...Substantial practical potential.”* (TiHi)

**(ii) Simplicity, efficiency, and broad applicability**

* *“Simple yet highly effective training-free ensembling method...substantial practical potential.”* (TiHi)
* *“Simple to implement, computationally efficient, and consistently effective.”* (aLzA)
* *“more efficient than prior works such as DeepEN.”* (TweT)

**(iii) Comprehensive empirical evaluation**

* *“The experiments are extremely well structured.”* (fhnh)
* *“Comprehensive experimental evaluations, demonstrating the effectiveness and broad applicability.”* (TiHi)
* *“The experimental evaluation is sufficiently thorough.”* (TweT)

---

# We conducted a detailed rebuttal that strengthened the paper.

Guided by the reviewers’ constructive feedback, we revised several parts of the main text to improve readability, discuss related work, and elaborate on technical details. Furthermore, we made new analyses and broader experiments as below:

*  **Theoretical Analysis** (Appendix C)
We added a detailed theoretical analysis of Token Translation, including a formal justification of its deterministic alignment properties and clarification of the underlying assumptions. These additions strengthen the theoretical grounding of the method and show that ToT provides causal alignment across heterogeneous token spaces.

* **Tokenizer Behavior Analysis** (Appendix B.4.1)
We analyzed tokenizer behavior by measuring token overlap across heterogeneous tokenizer pairs, quantifying how much of each vocabulary can be paired, and clarifying structural differences relevant to cross-model collaboration.

* **Token Alignment Analysis** (Appendix B.4.2 and B.4.3)
We added systematic analyses of alignment quality, covering forward and backward mapping behavior, prefix and superstring patterns, and semantic coherence based on embedding similarity. These studies clarify the structural and semantic properties of aligned token pairs.

* **Broader Task and Model Evaluation** (Appendix B.4.4 and B.4.5)
We expanded our empirical evaluation to include larger backbones such as Mixtral-8×7B and Qwen2.5-32B, and further highlighted our results on the generation tasks beyond multiple-choice questions, including summarization, open-domain QA, reasoning, translation, and code generation. These experiments show that the method remains robust and effective across diverse architectures and tasks.

* **Uncertainty-aware Weighting Analysis** (Appendix B.4.6)
We added an extended analysis of uncertainty-aware weighting, including top-k entropy ablations and comparisons with alternative schemes. These results show that our uncertainty signals provide meaningful improvements in ensemble performance.
---

# We appreciate the reviewers’ engagement, and we have responded thoroughly to all comments.

We provided point-by-point responses to every weakness and question raised by all four reviewers and incorporated corresponding revisions into the manuscript.

Among the reviewers who actively engaged during the previous discussion phase, Reviewer fhnh expressed strong support for acceptance, improving the scores for soundness, presentation, and novelty, and showed clear confidence in the strengthened contribution following our rebuttal.

Reviewer TweT also engaged thoroughly, and after our detailed clarifications on Assumption 2, EOS handling, and the theoretical scope of Token Translation, explicitly stated that they would like to take all the points and re-evaluate the paper accordingly.

Given the additional experiments, expanded analyses, and theoretical clarifications introduced during the rebuttal, we are confident that our responses address most, if not all, of the concerns raised by the reviewers, including those who have not yet had the opportunity to further engage in discussion.

---

We hope this summary is helpful in supporting the ongoing discussion and decision-making. All points presented here reflect objective information from our rebuttal and revisions. We sincerely thank the AC and all reviewers for their thoughtful evaluation and for helping us further improve this work.

Best,

#14478 Authors

---

### Meta-Review · Area_Chair_7CDT · 2026-01-07

**Summary:**

The initial ratings are 6, 6, 4, 4. This paper proposes token translation to address tokenizer mismatch in test-time LLM, which aligns heterogeneous token spaces by decoding tokens from a source tokenizer and re-encoding with a target tokenizer. The method uses bidirectional translation (prefix and superstring weighted mappings) and uncertainty-aware fusion to combine predictions from multiple models.

Strengths:
(1)The proposed method is practical potential, and it aligns the vocabularies of different models using only tokenization rules.
(2) The work conducts comprehensive experimental evaluations, demonstrating its effectiveness.

Weaknesses:
(1)The token-translation method is mainly heuristics and is not theoretically grounded compared to other approach, for example the character-ensemble baseline converts the every model to the same base character alphabet with statistical guarantees.
(2)The idea of using entropy for model combination is not new . It would be important to include a comparison against other methods with entropy-weighted ensembles.

**Reviewer Concerns:**

Some concerns of Reviewer aLzA and fhnh were addressed by the rebuttal, and Some main concerns of  Reviewer TweT and TiHi are still outstanding.

**Reviewer Scores:**

Reviewer aLzA and fhnh are possible to raise the scores.

---

### Decision · Program_Chairs · 2026-01-26

Reject